# WikiContradict: A Benchmark for Evaluating LLMs on Real-World Knowledge Conflicts from Wikipedia

**Yufang Hou**[1,2]**, Alessandra Pascale**[1]**, Javier Carnerero-Cano**[1]**, Tigran Tchrakian**[1]
**Radu Marinescu**[1]**, Elizabeth Daly**[1]**, Inkit Padhi**[3]**, Prasanna Sattigeri**[3]

[1] IBM Research Europe - Ireland
[2] IT:U Interdisciplinary Transformation University Austria
[3] IBM Research, Thomas J. Watson Research Center, Yorktown Heights, USA
{yhou|apascale|tigran|radu.marinescu|elizabeth.daly}@ie.ibm.com
{javier.cano|inkpad}@ibm.com, psattig@us.ibm.com

## Abstract

Retrieval-augmented generation (RAG) has emerged as a promising solution to mitigate the limitations of large language models (LLMs), such as hallucinations and outdated information. However, it remains unclear how LLMs handle knowledge conflicts arising from different augmented retrieved passages, especially when these passages originate from the same source and have equal trustworthiness. In this work, we conduct a comprehensive evaluation of LLM-generated answers to questions that have varying answers based on contradictory passages from Wikipedia, a dataset widely regarded as a high-quality pre-training resource for most LLMs. Specifically, we introduce `WikiContradict`, a benchmark consisting of 253 high-quality, human-annotated instances designed to assess the performance of LLMs in providing a complete perspective on conflicts from the retrieved documents, rather than choosing one answer over another, when augmented with retrieved passages containing real-world knowledge conflicts. We benchmark a diverse range of both closed and open-source LLMs under different QA scenarios, including RAG with a single passage, and RAG with 2 contradictory passages. Through rigorous human evaluations on a subset of `WikiContradict` instances involving 5 LLMs and over 3,500 judgements, we shed light on the behaviour and limitations of these models. For instance, when provided with two passages containing contradictory facts, all models struggle to generate answers that accurately reflect the conflicting nature of the context, especially for implicit conflicts requiring reasoning. Since human evaluation is costly, we also introduce an automated model that estimates LLM performance using a strong open-source language model, achieving an F-score of 0.8. Using this automated metric, we evaluate more than 1,500 answers from seven LLMs across all `WikiContradict` instances. To facilitate future work, we release `WikiContradict` at https://ibm.biz/wikicontradict.

## 1 Introduction

The advent of large language models (LLMs) [Brown et al., 2020] has revolutionized the field of Natural Language Processing (NLP), enabling unprecedented capabilities in text understanding and generation. However, static LLMs often suffer from outdated information and hallucinations. To mitigate these shortcomings, retrieval-augmented generation (RAG) techniques [Lewis et al., 2020] have been developed, which combine the strengths of LLMs with retrieved up-to-date information from external sources. While RAG frameworks have shown significant promise, it remains unclear how LLMs handle knowledge conflicts from different sources, including "*context-memory conflicts*", which refers to the retrieved context knowledge being in conflict with the parametric knowledge (memory) encapsulated within the LLM's parameters, and "*inter-context conflicts*", which refers to the

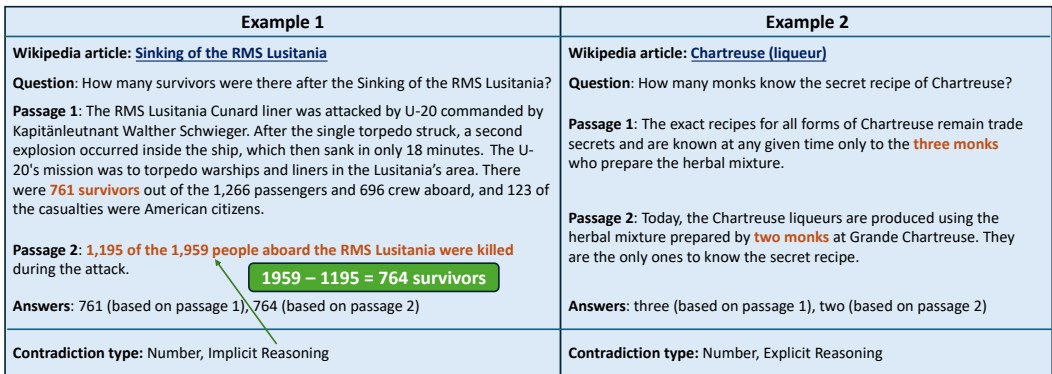

Figure 1: Example instances from `WikiContradict` with different contradiction types.

contradictions among the retrieved passages [Xu et al., 2024]. Most prior research on LLM knowledge conflicts has concentrated on "*context-memory conflicts*" and relied on artificially generated datasets, which employ various methods to create conflicting information. These approaches span from simple entity substitution, where an entity in a passage is replaced with another entity of the same type [Longpre et al., 2021], to more sophisticated techniques, such as instructing language models like ChatGPT to fabricate supporting evidence for counterfactual answers to a given question [Xie et al., 2024, Jin et al., 2024]. However, these artificially generated datasets primarily focus on explicit, surface-level contradictions, neglecting the complexity and nuance of real-world knowledge conflicts.

In this work, we focus on investigating the behaviors of LLMs when confronted with "***real-world inter-context conflicts***", where knowledge inconsistencies arise from the same or different retrieved passages that originate from a single trusted source (Wikipedia) and are considered equally credible. Specifically, we introduce `WikiContradict`, a benchmark consisting of 253 high-quality, human-annotated instances that cover different types of contradictions identified by Wikipedia editors and validated by us. Figure 1 presents two illustrative instances that demonstrate different types of contradictions. In particular, Example 1 requires implicit reasoning to detect the contradiction between Passage 1 and Passage 2 about the number of survivors from the RMS Lusitania sinking event, which requires calculating the number of survivors by subtracting 1,195 from 1,959 based on the information provided in Passage 2. This type of instance accounts for 36% of the instances in the `WikiContradict` dataset.

We evaluate the performance of various LLMs on `WikiContradict` by employing diverse prompt templates to assess their behaviour under different question answering (QA) scenarios, including RAG with a single context passage, and RAG with two contradictory passages. Our primary focus is on evaluating the ability of LLMs to provide a comprehensive and balanced perspective on conflicts by synthesizing information from the retrieved documents, rather than simply selecting one answer over another in scenarios where contradictory information is present. We then conduct a rigorous human evaluation to assess the correctness of the models' responses. Our human evaluation dataset comprises responses from 5 LLMs to 5 prompt templates, applied to 55 instances from the `WikiContradict` dataset, resulting in a total of 1,375 evaluation samples. Each sample is annotated by 2 authors of this paper, yielding 2,750 human judgements. The inter-annotator agreement, measured by Cohen's $\kappa$, ranges from 0.58 to 0.88 across different prompt templates, indicating moderate to substantial agreement. After resolving the annotation disagreements among annotators, our final human evaluation study dataset (`WikiContradict_HumanEval`) consists of 1,200 samples resulting from 5 LLMs' responses to 48 `WikiContradict` instances based on 5 prompt templates.

On `WikiContradict_HumanEval`, we observe that when instructing LLMs to generate answers to a given question based on the given context consisting of two contradicted passages, all models, including GPT-4, struggle to generate answers that accurately reflect the conflicting nature of the context, especially for implicit conflicts that require reasoning as illustrated in Figure 1, Example 1. Interestingly, we find that prompting LLMs to pay attention to contradictory context information improves their performance to correctly answering these questions. For instance, the top-performing model, Llama-3-70b-instruct, shows a remarkable increase from 10.4% to 43.8%. Furthermore, our analysis reveals that this improvement is largely driven by instances with explicit conflicts, as illustrated in Figure 1, Example 2. Finally, to facilitate future evaluations, we have developed

`WikiContradictEval`, a simple automatic evaluation method that leverages few-shot in-context learning to teach Llama-3-70b-instruct to judge model responses, which achieves an F-score of 0.8 on `WikiContradict_HumanEval` for evaluating LLM responses in the RAG setting with two contradictory passages.

In summary, our proposed `WikiContradict` benchmark poses a significant challenge for current LLMs, highlighting substantial opportunities for future improvement. We believe `WikiContradict` can serve as a valuable resource for the research community, facilitating the examination and tracking of LLMs' progress in handling real-world inter-context conflicts and deepening our understanding of their capabilities in these complex settings.

## 2   Related work

**Knowledge conflicts.**   Knowledge conflicts are commonly presented to LLMs and exploring the capability of the model to understand and manage them to ensure trustworthiness of the answer is gaining increasing interest in the community. While there exist three categories of conflicts: intra-memory, context-memory and inter-context [Xu et al., 2024] we focus our attention on the inter-context conflict. This type of contradiction has become of particular interest after the advent of RAG techniques. RAG has been proven to enhance LLMs' capabilities in dealing with hallucination and enrich LLMs' responses by integrating content from retrieved documents into the context [Lewis et al., 2021]. At the same time RAG can also introduce inconsistencies, as external documents may conflict with each other. In order to explore this phenomenon, previous research has relied on synthetically generated datasets containing conflicting statements [Chen et al., 2022, Wang et al., 2023, Li et al., 2024]. While they are used to evaluate and fine-tune existing models, these benchmarks fail to represent the complexity of real world conflicts [Xu et al., 2024]. We aim to shed light on the unexplored space of managing inter-context conflicts *in the wild*, starting from contradictions extracted and annotated from Wikipedia. Our aim is to assess how well LLMs perform in dealing with real-world scenarios, rather than with synthetically created conflicts, to better understand their behaviour and capability.

**LLMs evaluation benchmarks.**   Understanding adherence of LLMs to factual knowledge has gained increasing attention in recent years given the widespread use of these models. Hallucination detection and mitigation has been identified as a fundamental step to ensure transparency and trustworthiness of the models. In recent years a proliferation of benchmarks for evaluating factuality of LLMs has been observed with [Huang et al., 2023] presenting a survey of existing hallucination detection and mitigation approaches. They include many established benchmarks such as TruthfulQA [Lin et al., 2022], FreshQA [Vu et al., 2023], HaluEval [Li et al., 2023], HalluQA [Cheng et al., 2023], and FELM [Chen et al., 2023] that mainly focus on short-form answer evaluation where the knowledge of the LLM is tested in the form of a single factoid evaluated binary as true or false in adherence to the specific benchmark. More recent works [Wei et al., 2024, Min et al., 2023] cover long-form answer evaluation where the answer is decomposed into individual facts that are then independently evaluated. An ensemble metric is computed at the end to represent the overall evaluation score.

These previous works include the primary definition of truthfulness and factuality as a binary concept where the goal is to test the knowledge of the LLM in the form of a single (short-form) or multiple (long-form) factoids evaluated as **true or false** given the specific benchmark.

In this work we move away from a dualistic vision of the truth and we focus on cases where the answer to a question is not unique. We investigate how LLMs deal with real-world conflicting information where there exist different sources and possible answers considered equally trustworthy.

## 3   WikiContradict

In this section, we describe the process of leveraging contradiction tags from Wikipedia to develop `WikiContradict`, a QA-based benchmark consisting of 253 human-annotated instances that cover different types of real-world knowledge conflicts.

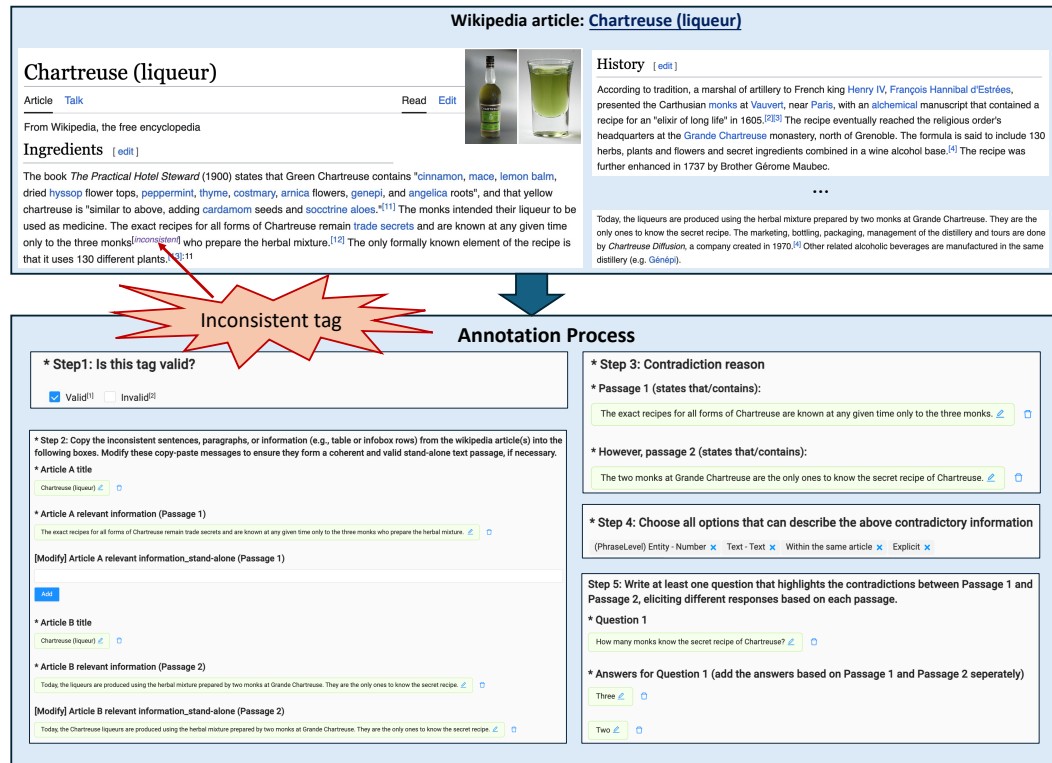

Figure 2: `WikiContradict` data annotation pipeline.

## 3.1 Data collection and processing

Although Wikipedia is widely regarded as a high-quality pre-training dataset for most LLMs, its content is not without flaws, including speculation, inconsistencies, and other content issues. To address these problems, Wikipedia editors use a wide range of maintenance tags to flag problematic content for improvement. However, these maintenance tags are typically removed when creating Wikipedia datasets for LLM pre-training, which results in content with various quality issues being included in the pre-training process.

In this work, we focus on three tags that indicate content inconsistencies: *inconsistent*, *self-contradictory*, and *contradict-other*. The first two tags denote contradictory statements within the same article, whereas the third tag highlights instances where the content of one article contradicts that of another article. In total, we collect around 1,200 articles that contain these tags through the Wikipedia maintenance category "*Wikipedia articles with content issues*"[1]. The upper portion of Figure 2 illustrates a Wikipedia article that has been flagged by an "*inconsistent*" tag with the comment "*contradictory number of monks*". The tagged paragraphs from these articles, together with the editors' comments whenever available, serve as the starting point for our annotations.

## 3.2 Data annotation

We developed an annotation interface using Label Studio[2], as shown in the lower portion of Figure 2. Given a content inconsistency tag provided by Wikipedia editors, our annotators first need to verify whether the tag is valid by checking the relevant article content, the editor's comment, as well as the information in the edit history and the article's talk page if necessary.

For the verified valid tags, we instruct annotators to extract the two contradictory paragraphs/sentences from the original article(s), slightly modifying them as needed to create stand-alone passages. Such modifications normally require the annotators to resolve anaphoric references (e.g., *She*) in the first

---

[1]https://en.wikipedia.org/wiki/Category:Wikipedia_articles_with_content_issues. The dataset is collected on Feburary 26, 2024.

[2]https://labelstud.io/

sentence of each passage. Next, the annotators should provide a brief explanatory text to clarify the contradictions between the two passages and categorize them using a pre-defined taxonomy, such as *Date, Number, Explicit, Implicit*. Briefly speaking, our contradiction taxonomy consists of four dimensions, which provide a comprehensive framework for categorizing and analyzing contradictions:

**Semantic Type (I)**: This dimension focuses on the fine-grained semantics of the contradiction, specifically the type of entity involved. We adapt OntoNotes named entity type definitions to describe the different types of contradicted entities, such as Date/Time, Location, Number, and others.

**Modality (II)**: This dimension examines the modality of the contradiction, describing the source of the information in both passages. This includes whether the information comes from a piece of text, a table, an infobox, or other sources.

**Origin (III)**: This dimension determines whether the contradictions originate from the same Wikipedia article or from different articles.

**Reasoning Type (IV)**: This dimension focuses on the reasoning aspect of the contradiction, distinguishing between explicit contradictions (where the contradiction is clearly stated) and implicit contradictions (where the contradiction is implied or requires inference).

Finally, the annotators must craft at least one question that effectively highlights the contradictions between the two passages, eliciting different answers depending on which passage is referenced. The two examples from Figure 1 illustrate our annotation results. On average, annotators spent around 30 minutes to annotate a tag; longer times are required to annotate tags related to implicit conflicts, especially for cases in which the reasons of inconsistency are not explicitly mentioned in the comments from Wikipedia editors. More details about the pre-defined contradiction taxonomy and the full annotation guideline can be found in Appendix A.

### 3.3  Data statistics

Using the annotation pipeline outlined in the previous section, the authors annotated the collected Wikipedia articles marked with inconsistency tags, yielding a total of 253 annotated instances. Each instance comprises a question, two contradictory passages, and two distinct answers, each derived from one of the passages. Table 1 shows an overview of the dataset statistics. Notably, among all annotated instances, approximately 61% of questions are categorized as wh-questions seeking specific information. Furthermore, a significant proportion of instances (36%) contain implicit contradictions.

Table 1: `WikiContradict` dataset.

| | |
|---|---|
| Wikipedia tags | 261 |
| Verified valid tags | 130 |
| Annotated instances | 253 |
| *Question type* | |
| Yes/No questions | 133 (53%) |
| Wh-questions | 120 (47%) |
| *Contradiction type* | |
| Explicit | 161 (64%) |
| Implicit | 92 (36%) |

### 3.4  Evaluation

To investigate how LLMs respond to real-world inter-context conflicts, we develop five prompt templates to evaluate their performance under different question-answering (QA) scenarios. As illustrated in Figure 3, for each annotated instance from the `WikiContradict` dataset, we generate five question prompts based on these pre-defined templates. Specifically, template 1 evaluates a model's internal knowledge, while templates 2 and 3 examine its performance in the RAG setting with a single retrieved passage. Templates 4 and 5, on the other hand, assess a model's ability to handle QA in the RAG setting with two contradictory passages that can lead to different answers.

To evaluate LLMs' responses to these question prompts, we follow the relaxed evaluation mode from FreshLLM [Vu et al., 2023] by allowing additional hallucinated or inaccurate information as long as the primary answer is accurate and any additional information does not contradict with the primary answer. More specifically, each response is evaluated as "*correct*", "*partially correct*", or "*incorrect*":

- "*Correct*" if the response accurately matches all the answers in the annotated answer list. For prompt templates 4 and 5, the response should identify and contain the contradictory answers that reflect the heterogeneous nature of the context. Additionally, the correct response should not

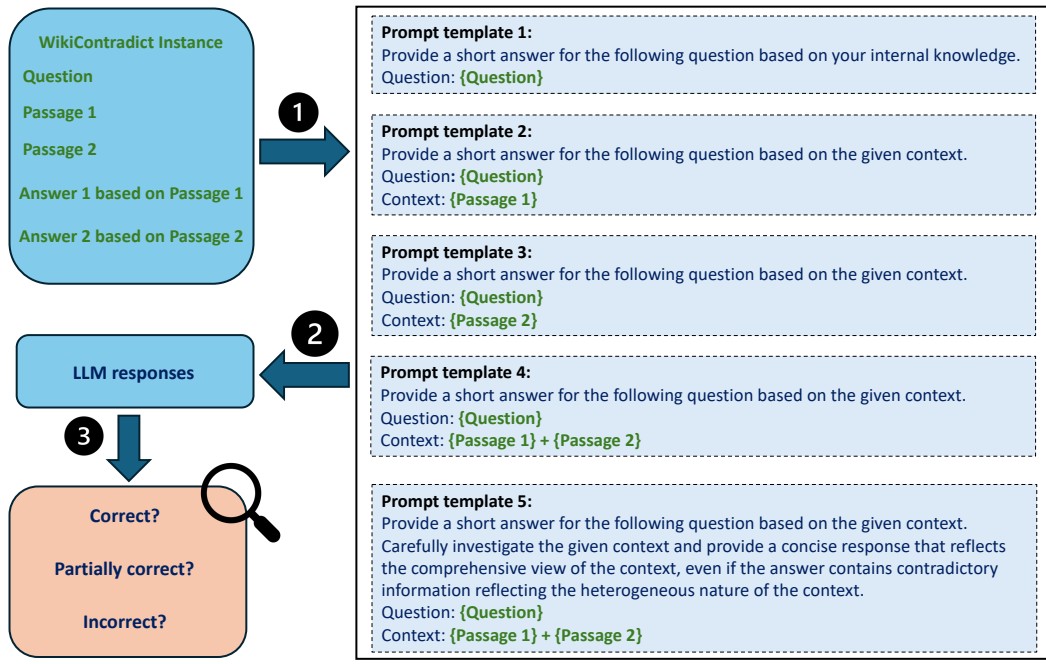

Figure 3: WikiContradict evaluation.

indicate a preference for one answer over another, and it should not combine two different correct answers without indicating the contradictory nature of these answers.

- "*Partially correct*" applies to prompt templates 1, 4, and 5; it means that the response only matches one of the answers in the annotated answer list, or the response matches all the answers in the correct answer list but indicates a preference for one answer over another.

- "*Incorrect*" if the response does not match any of the annotated answers, or the response merely combines two contradictory answers from the annotated answer list and indicates that both are possible at the same time without indicating the contradictory nature of the two context passages.

Following the criteria described above, for the question from Example 2 of Figure 1, an LLM response to prompt template 4, "*According to the context, the number of monks who know the secret recipe of Chartreuse is either three or two. The first statement suggests that three monks know the recipe, while the second statement contradicts this, stating that only two monks at Grande Chartreuse know the secret recipe.*" is a **correct** response. In contrast, the LLM response to prompt template 3, "*Two monks know the secret recipe of Chartreuse.*" is **partially correct**, as it only captures one aspect of the contradictory information presented in the context.

## 4   Human evaluation: LLMs are struggling on WikiContradict

### 4.1   Main evaluation with inter-context conflicts

To understand LLMs' behavior when faced with real-world inter-context conflicts, we use WikiContradict to benchmark a list of LLMs with the evaluation protocol described in Section 3.4[3]. The authors independently evaluated a subset of answers, comprising 1,375 responses from five LLMs based on the five prompt templates, as shown in Figure 3, for 55 instances. Each response is assessed by two authors of this paper, yielding a total of 2,750 human judgements. The inter-annotator agreements, as measured by Cohen's kappa $\kappa$, were moderate to substantial, with values of 0.58 for prompt template 3, 0.67 for prompt template 4, 0.84 for prompt template 0, 0.85 for prompt template 2, and 0.88 for prompt template 1. After resolving the annotation disagreements among annotators,

---

[3]In our experiments, decoding temperature=0, maximum output length=250 tokens.

Table 2: Human evaluation results on `WikiContradict_HumanEval`. "C", "PC" and "IC" stand for "*Correct*", "*Partially correct*", "*Incorrect*", respectively. "all", "exp", and "imp" represent for instance types: all instances, instances with explicit conflicts, and instances with implicit conflicts. The numbers represent the ratio of responses from each LLM that were assessed as "*Correct*, "*Partially correct*, or "*Incorrect* for each instance type under a prompt template. The bold numbers highlight the best models that correctly answer questions for each type and prompt template.

| | Mistral-7b-inst | | | Mixtral-8x7b-inst | | | Llama-2-70b-chat | | | Llama-3-70b-inst | | | GPT-4 | | |
|---|---|---|---|---|---|---|---|---|---|---|---|---|---|---|---|
| | **all** | **exp** | **imp** | **all** | **exp** | **imp** | **all** | **exp** | **imp** | **all** | **exp** | **imp** | **all** | **exp** | **imp** |
| **Prompt Template 1** | | | | | | | | | | | | | | | |
| C | **4.2** | **6.7** | 0.0 | 2.1 | 0.0 | 5.9 | 0.0 | 0.0 | 0.0 | **4.2** | 0.0 | **11.8** | 2.1 | 0.0 | 5.9 |
| PC | 33.3 | 23.3 | 47.1 | 52.1 | 43.3 | 64.7 | 54.2 | 43.3 | 70.6 | 52.1 | 46.7 | 58.8 | 58.3 | 53.3 | 64.7 |
| IC | 62.5 | 70.0 | 52.9 | 45.8 | 56.7 | 29.4 | 45.8 | 56.7 | 29.4 | 43.8 | 53.3 | 29.4 | 39.6 | 46.7 | 29.4 |
| **Prompt Template 2** | | | | | | | | | | | | | | | |
| C | 92.7 | - | - | **97.6** | - | - | 87.8 | - | - | 95.1 | - | - | **97.6** | - | - |
| IC | 7.3 | - | - | 2.4 | - | - | 12.2 | - | - | 4.9 | - | - | 2.4 | - | - |
| **Prompt Template 3** | | | | | | | | | | | | | | | |
| C | 82.9 | - | - | **92.7** | - | - | 90.2 | - | - | **92.7** | - | - | 87.8 | - | - |
| IC | 17.1 | - | - | 7.3 | - | - | 9.8 | - | - | 7.3 | - | - | 12.2 | - | - |
| **Prompt Template 4** | | | | | | | | | | | | | | | |
| C | 2.1 | 3.3 | 0.0 | 4.2 | 3.3 | 5.9 | 4.2 | 3.3 | 5.9 | **10.4** | **13.3** | 5.9 | 6.3 | 3.3 | **11.8** |
| PC | 87.5 | 86.7 | 88.2 | 91.7 | 93.3 | 88.2 | 93.8 | 96.7 | 88.2 | 81.3 | 80.0 | 82.4 | 85.4 | 96.7 | 64.7 |
| IC | 10.4 | 10.0 | 11.8 | 4.2 | 3.3 | 5.9 | 2.1 | 0.0 | 5.9 | 8.3 | 6.7 | 11.8 | 8.3 | 0.0 | 23.5 |
| **Prompt Template 5** | | | | | | | | | | | | | | | |
| C | 20.8 | 26.7 | 11.8 | 14.6 | 16.7 | 11.8 | 22.9 | 26.7 | **17.6** | **43.8** | **60.0** | **17.6** | 10.4 | 10.0 | 11.8 |
| PC | 70.8 | 63.3 | 82.4 | 83.3 | 83.3 | 82.4 | 68.8 | 63.3 | 76.5 | 45.8 | 26.7 | 76.5 | 81.3 | 90.0 | 64.7 |
| IC | 8.3 | 10.0 | 5.9 | 2.1 | 0.0 | 5.9 | 8.3 | 10.0 | 5.9 | 10.4 | 13.3 | 5.9 | 8.3 | 0.0 | 23.5 |

our final human evaluation study dataset (`WikiContradict_HuamEval`) consists of 1,200 samples resulting from five LLMs' responses to 48 WikiContradict instances based on five prompt templates[4].

Table 2 presents the results of `WikiContradict_HumanEval` for 5 LLMs: *Mistral-7b-instruct*, *Mixtral-8x7b-instruct*, *Llama2-2-70b-chat*, *Llama3-70b-instruct*, and *GPT-4-turbo-2024-04-09*. The table provides a detailed breakdown of response accuracy for each LLM, categorized into three types: *correct*, *partially correct* (applicable to prompt templates 1, 4, and 5), and *incorrect*. We further distinguish between instances with explicit conflicts (30 instances) and implicit conflicts (18 instances) to provide a more nuanced understanding of the LLMs' performance. Below we summarise a few key observations on `WikiContradict_HumanEval`.

**Prompt template 1: There is a significant overlap between the internal knowledge of LLMs and the content of Wikipedia.** As expected, the portion of correct or partially correct answers based on their internal knowledge, ranges from 37.5% (*Mistral-7b-inst*) to 60.4% (*GPT-4*).

**Prompt template 2 and 3: When tasked with answering questions based on a provided context, LLMs are generally capable of generating correct responses for the majority of instances, as long as the context does not contain conflicting information.** However, we observe that *GPT-4* exhibits a "stubborn" behavior, particularly with prompt template 3. It often relies on its internal knowledge, which may not align with the given context, resulting in a lower accuracy of 87.8% compared to *Mixtral-8X7b-inst* and *Llama3-3-70b-inst*, which perform better in this scenario.

**Prompts template 4 and 5: LLMs often struggle to provide correct answers when the context contains conflicting information.** Typically, the models rely on a single passage to inform their response, disregarding the other passage. Notably, some models attempt to reconcile the conflicting information by providing both answers, but then proceed to explain why one of them is incorrect. This phenomenon is particularly pronounced in the *Mistral-7b-inst* model.

**Prompts template 4 vs. 5: LLMs can improve their performance in providing correct answers when explicitly instructed to consider conflicting information within the given context.** Notably, *Llama-3-70b-inst* exhibits the most substantial improvement, jumping from 10.4% to 43.8%. In contrast, *GPT-4* demonstrates the smallest improvement, increasing from 6.3% to 10.4%, which is likely attributed to its previously observed stubborn behavior in prompt template 3.

---

[4]We exclude 7 instances from our human evaluation study due to ambiguity in their inconsistencies, which are not crystal clear. These instances are also excluded from the final `WikiContradict` dataset.

Table 3: Additional human evaluation results on `WikiContradict_HumanEval_Control`. "C", "PC" and "IC" stand for "*Correct*", "*Partially correct*", "*Incorrect*", respectively. "all", "exp", and "imp" represent for instance types: all instances, instances with explicit conflicts, and instances with implicit conflicts. The numbers represent the ratio of responses from each LLM that were assessed as "*Correct*, "*Partially correct*, or "*Incorrect* for each instance type under a prompt template. The bold numbers highlight the best models that correctly answer questions for each type and prompt template.

| | Mistral-7b-inst | | | Mixtral-8x7b-inst | | | Llama-2-70b-chat | | | Llama-3-70b-inst | | | GPT-4 | | |
|---|---|---|---|---|---|---|---|---|---|---|---|---|---|---|---|
| | **all** | **exp** | **imp** | **all** | **exp** | **imp** | **all** | **exp** | **imp** | **all** | **exp** | **imp** | **all** | **exp** | **imp** |
| **Prompt Template 5** | | | | | | | | | | | | | | | |
| C | 20.8 | 26.7 | 11.8 | 14.6 | 16.7 | 11.8 | 22.9 | 26.7 | **17.6** | **43.8** | **60.0** | **17.6** | 10.4 | 10.0 | 11.8 |
| PC | 70.8 | 63.3 | 82.4 | 83.3 | 83.3 | 82.4 | 68.8 | 63.3 | 76.5 | 45.8 | 26.7 | 76.5 | 81.3 | 90.0 | 64.7 |
| IC | 8.3 | 10.0 | 5.9 | 2.1 | 0.0 | 5.9 | 8.3 | 10.0 | 5.9 | 10.4 | 13.3 | 5.9 | 8.3 | 0.0 | 23.5 |
| **Prompt Template 5.1** | | | | | | | | | | | | | | | |
| C | 22.9 | 30.0 | 11.8 | 14.6 | 16.7 | 5.9 | 17.0 | 27.6 | 0.0 | **54.2** | **70.0** | **23.5** | 12.5 | 16.7 | 5.9 |
| PC | 72.9 | 66.7 | 82.4 | 70.8 | 73.3 | 70.6 | 80.9 | 72.4 | 94.1 | 45.8 | 30.0 | 76.5 | 81.3 | 83.3 | 76.5 |
| IC | 4.2 | 3.3 | 5.9 | 14.6 | 10.0 | 23.5 | 2.1 | 0.0 | 5.9 | 0.0 | 0.0 | 0.0 | 6.3 | 0.0 | 17.6 |
| **Prompt Template 5.2** | | | | | | | | | | | | | | | |
| C | 35.4 | 40.0 | 29.4 | 4.2 | 6.7 | 0.0 | 76.6 | 89.7 | 52.9 | 77.1 | 93.3 | 47.1 | **89.6** | **96.7** | **76.5** |
| PC | 14.6 | 20.0 | 0.0 | 6.3 | 6.7 | 5.9 | 17.0 | 10.3 | 29.4 | 6.3 | 3.3 | 11.8 | 2.1 | 3.3 | 0.0 |
| IC | 50.0 | 40.0 | 70.6 | 72.9 | 76.7 | 70.6 | 6.4 | 0.0 | 17.6 | 14.6 | 3.3 | 35.3 | 8.3 | 0.0 | 23.5 |
| **Prompt Template 5.3** | | | | | | | | | | | | | | | |
| C | 87.5 | 93.3 | 82.4 | 85.4 | 83.3 | **88.2** | 87.2 | 96.6 | 76.5 | **89.6** | **96.7** | 82.4 | 87.5 | **96.7** | 70.6 |
| PC | 6.3 | 3.3 | 5.9 | 4.2 | 6.7 | 0.0 | 2.1 | 0.0 | 5.9 | 2.1 | 0.0 | 5.9 | 2.1 | 0.0 | 5.9 |
| IC | 6.3 | 3.3 | 11.8 | 10.4 | 10.0 | 11.8 | 10.6 | 3.4 | 17.6 | 8.3 | 3.3 | 11.8 | 10.4 | 3.3 | 23.5 |
| **Prompt Template 5.4** | | | | | | | | | | | | | | | |
| C | 66.7 | 70.0 | 64.7 | **83.3** | **83.3** | **82.4** | 12.8 | 6.9 | 23.5 | 47.9 | 50.0 | 41.2 | 43.8 | 50.0 | 35.3 |
| PC | 4.2 | 6.7 | 0.0 | 2.1 | 3.3 | 0.0 | 23.4 | 24.1 | 17.6 | 4.2 | 3.3 | 5.9 | 14.6 | 10.0 | 23.5 |
| IC | 29.2 | 23.3 | 35.3 | 2.1 | 3.3 | 0.0 | 63.8 | 69.0 | 58.8 | 45.8 | 43.3 | 52.9 | 41.7 | 40.0 | 41.2 |

**Explicit conflicts vs. implicit conflicts: When explicitly instructed to consider conflicting information within the given context, LLMs' performance in providing correct answers improves in particular in cases where conflicts are explicitly stated.** For instance, for *Llama-3-70b-inst*, the performance on explicit conflicts instances jumps from 13.3% to 60.0%, while the performance on implicit conflicts instances improves from 5.9% to 17.6%.

**More insights on errors for partially correct and incorrect:** In the RAG setup (Table 2, Prompt Template 2 - 5), models' answers rarely fall into the "*incorrect*" category. When such cases happen, the model's answer acknowledge coexistence of two facts which are not logically correct, such as stating a person was born on two dates (e.g., *According to the provided context, Paul McCole was born on 1 February 1972 and 10 February 1972*). In contrast, when instructed to answer questions based on their internal knowledge without providing any context (Table 2, Prompt Template 1), the ratio of "*incorrect*" answers increases significantly. This is likely due to the fact that LLMs either memorize a different answer from another source other than Wikipedia during pre-training or hallucinate the answer. Regarding partially correct answers, most LLMs tend to produce a single answer based on only one given context, neglecting the other. However, some models, such as *Mistral-7b-inst*, attempt to reconcile the conflicting information by providing both answers and then explaining why one of them is incorrect. This phenomenon is particularly pronounced in the *Mistral-7b-inst* model.

## 4.2 Additional evaluation on the control setups

We conducted additional human evaluation studies on 48 instances from `WikiContradict_HumanEval` using four variations of prompt template 5: prompt template 5.1 swaps the positions of passage 1 and passage 2 from the original template 5; prompt template 5.2 instructs LLMs to identify any contradictions in the given context with respect to the question; prompt template 5.3 provides LLMs with manually revised passage 1 and passage 2, where contradictions with respect to the question were resolved; prompt template 5.4 tasks LLMs with detecting any contradictions in the revised consistent context from template 5.3 with respect to the question. In this experiment, each response is assessed by a single human annotator. Please refer to Appendix B for more details on the prompt templates 5.1 - 5.4. Table 3 presents the results of the human evaluation of five LLMs on five prompt templates (5, 5.1, 5.2, 5.3, and 5.4), yielding the following key insights:

Table 4: Judge LLM results for prompt template 5 on `WikiContradict_HumanEval`. GTP-4 and GPT-4o represent "gpt-4-turbo-2024-04-09" and "gpt-4o-2024-05-13", respectively.

| | Acc | Macro-F | *Correct* (P/R/F) | *Partially correct* (P/R/F) | *Incorrect* (P/R/F) |
|---|---|---|---|---|---|
| Mixtral-8x7b-inst | 26.2 | 19.7 | 10.5 / 22.0 / 14.2 | 66.2 / 30.0 / 41.3 | 2.9 / 5.0 / 3.7 |
| Llama-3-70b-inst | 85.4 | 74.4 | 72.6 / 90.0 / 80.4 | 96.1 / 87.6 / 91.7 | 47.8 / 55.0 / 51.2 |
| GPT-4 | **86.7** | **76.1** | **73.4 / 94.0 / 82.5** | **96.8 / 88.2 / 92.3** | **52.4 / 55.0 / 53.7** |
| GPT-4o | 83.3 | 71.3 | 74.5 / 76.0 / 75.2 | 94.9 / 88.2 / 91.5 | 38.7 / 60.0 / 47.1 |

Table 5: Comparing human judgement with Llama3-70b-instruct judgement for prompt template 5 for five testing LLMs on `WikiContradict_HumanEval`.

| | Human judgement | | | LLM judgement | | |
|---|---|---|---|---|---|---|
| | *Correct* | *Partially correct* | *Incorrect* | *Correct* | *Partially correct* | *Incorrect* |
| Mistral-7b-inst | 20.8 | 70.8 | 8.3 | 39.6 | 54.2 | 6.2 |
| Mixtral-8x7b-inst | 14.6 | 83.3 | 2.1 | 16.7 | 77.1 | 6.2 |
| Llama-2-70b-chat | 22.9 | 68.8 | 8.3 | 22.9 | 64.6 | 12.5 |
| Llama-3-70b-inst | 43.8 | 45.8 | 10.4 | 43.8 | 41.7 | 14.6 |
| GPT-4 | 10.4 | 81.3 | 8.3 | 8.3 | 81.2 | 10.4 |

1) **No position bias**: the results for prompt templates 5 and 5.1 are similar for all LLMs;

2) **Contradiction detection**: for prompt template 5.2, all LLMs perform better in detecting contradictions in the given context compared to generating correct answers for prompt template 5. *GPT-4* and *Llama-3-70b-instruct* are the top performers, with *GPT-4* detecting contradictions and providing reasonable explanations in around 88% of cases, followed by *Llama-3-70b-instruct* with 77%;

3) **Conflict-free context**: for prompt template 5.3, all models demonstrate high performance in correctly answering questions based on the given context (with correct response rates ranging from 85% to 90%), where there are no conflicts between passage 1 and passage 2. This is consistent with the observations for prompt templates 2 and 3 in the previous section;

4) **Difficulty in detecting conflict-free contexts**: for prompt template 5.4, all models struggle to detect when there are no contradictions in the given context, performing worse than when generating correct answers for prompt template 5.3. *GPT-4* and *Llama-3-70b-instruct* are the worst performers, only confirming 44% and 48% of instances as conflict-free, and "making up" reasons to explain why the two passages conflict in the remaining instances.

# 5 Automatic evaluation: WikiContradictEval

Since human evaluation is costly and time-consuming, to facilitate future evaluations, we have developed `WikiContradictEval`, a simple automatic evaluation method that leverages few-shot in-context learning to teach LLMs to judge model responses for prompt template 5, which is aligned with the central focus of `WikiContradict` benchmark to evaluate LLMs on real-world inter-context conflicts. Table 4 reports the results of different judge LLMs on the testing dataset that consists of 240 responses to prompt template 5 from five testing LLMs in `WikiContradict_HumanEval`. Among the four judge LLMs evaluated, the top-performing model, *GPT-4*, achieves an F-score of 82.5 in accurately identifying correct responses, with a precision score of 73.4 and a recall score of 94.0. The best open-source model, *Llama-3-70b-inst*, closely follows, with an overall F-score of 80.4 in identifying correct responses. For a detailed description of the judge LLM prompt, please see Appendix C.

Table 5 presents a comparison of human judgments with the assessments generated by the *Llama-3-70b-inst* judge model for prompt template 5 on the `WikiContradict_HumanEval` dataset. Note that our LLM judge model is generally well-aligned with human judgment in identifying correct responses for most LLMs, with one notable exception. The *Mistral-7b-instruct* model often attempts to reconcile conflicting information by providing both answers, which our LLM judge model mistakenly views as valid responses, as a result, it overestimates the correct responses.

Finally we apply the best open-source judge model, *Llama-3-70b*, to assess a list of seven LLMs based on prompt template 5 on all 253 instances from the `WikiContradict` dataset. Table 6 shows the evaluated results for each LLM. Among all testing models, *Mistral-7b-inst* and *Llama-3-70b-inst*

Table 6: *Llama-3-70b* judge results on `WikiContradict` based on prompt template 5. "C", "PC", and "IC" stand for "*Correct*", "*Partially correct*", "*Incorrect*", respectively.

| | All instances | | | Explicit contradictions | | | Implicit contradictions | | |
|---|---|---|---|---|---|---|---|---|---|
| | C | PC | IC | C | PC | IC | C | PC | IC |
| Mistral-7b-inst | 50.6 | 39.9 | 5.9 | 57.8 | 36 | 3.1 | 38.0 | 46.7 | 10.9 |
| Llama-3-70b-inst | 45.8 | 38.7 | 13.8 | 55.9 | 28.6 | 14.3 | 28.3 | 56.5 | 13.0 |
| Llama-3-8b-inst | 37.9 | 53 | 6.7 | 47.8 | 46.6 | 5.0 | 20.7 | 64.1 | 9.8 |
| Mixtral-8x7b-inst | 37.9 | 52.2 | 6.3 | 43.5 | 48.4 | 3.7 | 28.3 | 58.7 | 10.9 |
| GPT-4 | 15.0 | 73.5 | 11.5 | 19.3 | 70.8 | 9.9 | 7.6 | 78.3 | 14.1 |
| Llama-2-13b-chat | 11.5 | 73.5 | 9.5 | 12.4 | 74.5 | 7.5 | 9.8 | 71.7 | 13.0 |
| Flan-ul2 | 1.2 | 90.1 | 7.9 | 0 | 92.5 | 6.8 | 3.3 | 85.9 | 9.8 |

are the top performers in correctly answering questions, with correct response rates of 50.6% and 45.8%, respectively. It is noteworthy that, based on the above analysis, there is a high probability that the judge LLM overestimates the performance of *Mistral-7b-inst*, suggesting a potential bias in its evaluation. In addition, all models except Flan-ul2 have higher correct response rates for instances with explicit contradictions compared to instances with implicit contradictions, which is aligned with the observation in our human evaluation study (Section 4).

# 6   Limitation and future work

We acknowledge several limitations of the current benchmark. Firstly, the benchmark presented in this paper is restricted to English language instances, which may not generalize to other languages due to the nuanced nature of contradictory statements. Secondly, the reliance on Wikipedia contradictory tags may introduce bias towards specific types of contradictory statements, limiting the benchmark's representativeness. Future work can leverage automatic methods to detect contradictory statements from Wikipedia (e.g., [Hsu et al., 2021]) to expand the dataset. Third, the benchmark only covers text-text contradictions, whereas contradictions can occur across modalities, such as between text and images, as demonstrated by some Wikipedia contradictory tags that capture discrepancies between textual descriptions and visual content. Future work will aim to incorporate coverage across different modalities. Lastly, the proper handling of contradictions remains a challenging question. While prior studies have utilized contradictions to detect misinformation or disinformation [Glockner et al., 2022, 2024a,b, Weller et al., 2024], our current approach outlines the contradiction in the response, assuming all retrieved passages come from the same credible resource. An alternative approach would be to cross-check each answer separately with other credible resources and provide a confidence score for the correctness of each individual answer.

# 7   Conclusion

Unlike most previous work on LLM knowledge conflicts within RAG frameworks, which focus on "*context-memory conflicts*", our focus is on "*real-world inter-context conflicts*". Within this setting we introduced the `WikiContradict` benchmark, which consists of 253 human-annotated instances covering different types of contradictions identified by Wikipedia editors. Our annotation of these instances, which includes isolation of relevant conflicting passages, resolution of anaphora and the creation of at least one question that highlights the contradictions, results in a benchmark that can effectively evaluate the capacity of LLMs to manage and reason over knowledge conflicts. This capacity was highlighted by the results of the evaluation of LLMs on the benchmark, in which for each contradictory instance, the LLMs were given different prompts, each one containing a different instruction and/or different conflicting information contained within the instance. Our experiments show that LLMs often struggle to correctly identify and manage real-world inter-context conflicts, indicating the usefulness of the benchmark and the need for further research in this direction.

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

# A   Annotation Guideline

Here we include the details about the pre-defined contradiction taxonomy and the full annotation guideline. The annotation interface was developed using Label Studio[5].

---

[5]https://labelstud.io/

# 1. Setup the Label Studio environment

1) install Label Studio locally
   Github: https://github.com/HumanSignal/label-studio
   Here's the command I used to install it in my laptop (make sure python>=3.8):

   ```
   conda create --name label-studio
   conda activate label-studio
   conda install psycopg2
   pip install label-studio
   ```

   About LS version: I'm using Label Studio 1.8.2. Normally all new versions later than 1.8.2 should work, I recommend using google Chrome to do annotation.

2) run "label-studio" to start the server at http://localhost:8080
3) create an account and log in
4) create a new project called "WikipediaContradict" or any other names

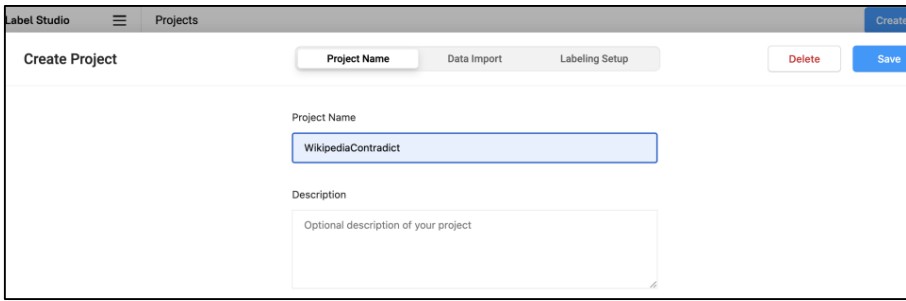

Fig 1: Create a new project

5) import the annotation tasks into the platform: first, click "upload files" and choose "AnnotationFilesSplit_new/inconsistentArticleTags_all_X.json"; next, click "save" to finish importing the 170 annotation tasks. Please refer to the excel file (AnnotationAssignment.xlsx) to identify which file you need to import.

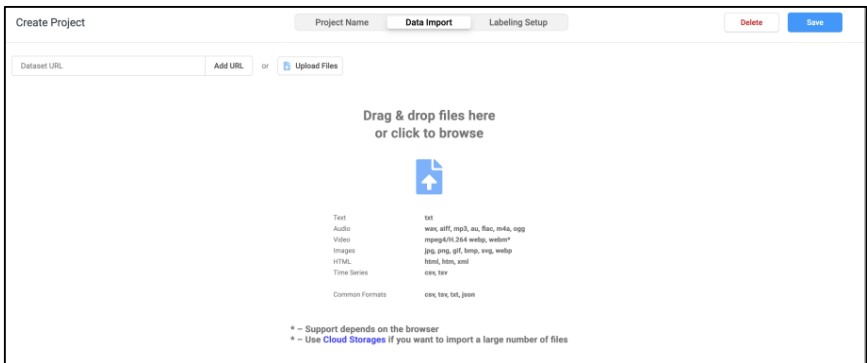

Fig 2: Import annotated data

6) setup the annotation UI:
   a) click "settings" on the upper right corner:

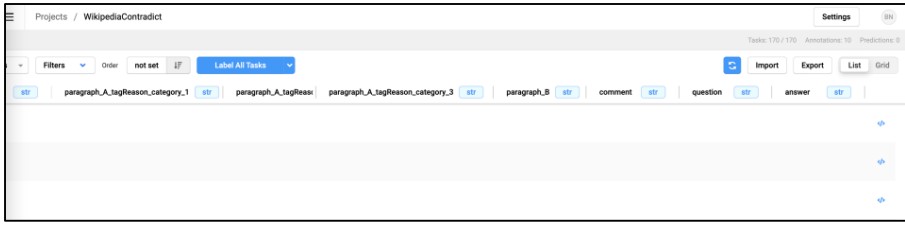

Fig 3: Setup the annotation UI

b) click "labelling interface", copy-paste the following UI code into the box (or the code from the "WikipediaContradiction_LabelSstudioUI"), then click "save". After this, you can start annotating by clicking "Label All Tasks" or any task in the panel.

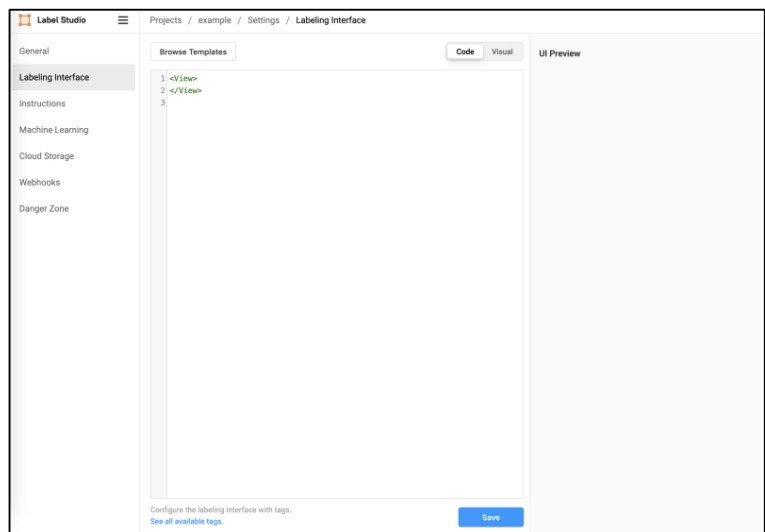

Fig 4: Setup the annotation UI - Continue

```
<View>
<!-- <header name="articletitle" value="Wikipedia article: $title"/> -->
 <HyperText clickableLinks="true" name="articlelink" inline="true" target="_blank" value="">
   <h2><a target="_blank" href="$url">Wikipedia article: $title</a></h2>
 </HyperText>

 <View style="box-shadow: 2px 2px 5px #999;        padding: 20px; margin-top: 2em;        border-radius: 5px;">
 <Header value="Inconsistence or contradictory tag"/>
 <Text name="wikitag" value="$paragraph_A"/>
 </View>

 <View style="box-shadow: 2px 2px 5px #999;        padding: 20px; margin-top: 2em;        border-radius: 5px;">
 <Header value="* Step1: Is this tag valid?"/>
 <Choices name="wikitag_label_valid" toName="wikitag" choice="single" showInLine="true">
   <Choice value="Valid"/>
   <Choice value="Invalid"/>
 </Choices>
 <Header value="Additional comment" />
   <TextArea name="valid_comment" toName="wikitag"
       showSubmitButton="true" maxSubmissions="1" editable="true"
       required="false" />

</View>

<View visibleWhen="choice-selected" whenTagName="wikitag_label_valid" whenChoiceValue="Valid" style="box-shadow: 2px
2px 5px #999;        padding: 20px; margin-top: 2em;  border-radius: 5px; ">
```

```
<Header value="* Step 2: Copy the inconsistent sentences, paragraphs, or information (e.g., table or infobox rows) from the
wikipedia article(s) into the following boxes. Modify these copy-paste messages to ensure they form a coherent and valid stand-
alone text passage, if necessary."/>
<Header value="* Article A title" />
<TextArea name="paragraphA_article" toName="wikitag"
        showSubmitButton="true" maxSubmissions="1" editable="true"
        required="true" />

<Header value="* Article A relevant information (Passage 1)" />
  <TextArea name="paragraphA_information" toName="wikitag"
        showSubmitButton="true" maxSubmissions="1" editable="true"
        required="true" />
<Header value="[Modify] Article A relevant information_stand-alone (Passage 1)" />
  <TextArea name="paragraphA_information_standalone" toName="wikitag"
        showSubmitButton="true" maxSubmissions="1" editable="true"
        required="false" />

<Header value="* Article B title" />

  <TextArea name="paragraphB_article" toName="wikitag"
        showSubmitButton="true" maxSubmissions="1" editable="true"
        required="true" />

<Header value="* Article B relevant information (Passage 2)" />
  <TextArea name="paragraphB_information" toName="wikitag"
        showSubmitButton="true" maxSubmissions="1" editable="true"
        required="true" />
<Header value="[Modify] Article B relevant information_stand-alone (Passage 2)" />
  <TextArea name="paragraphB_information_standalone" toName="wikitag"
        showSubmitButton="true" maxSubmissions="1" editable="true"
        required="false" />

  <Header value="* Are Passage 1 and Passage 2 the same?"/>
  <Choices name="wikitag_label_samepassage" toName="wikitag" choice="single" showInLine="true">
    <Choice value="Same"/>
    <Choice value="Different" selected="true"/>
  </Choices>
</View>

<View visibleWhen="choice-selected" whenTagName="wikitag_label_valid" whenChoiceValue="Valid" style="box-shadow: 2px
2px 5px #999;          padding: 20px; margin-top: 2em;  border-radius: 5px; ">
<Header value="* Step 3: Contradiction reason" />
<Header size = "8">* Passage 1 (states that/contains):  </Header> <TextArea name="relevantInfo_comment_A" toName="wikitag"
        showSubmitButton="true" maxSubmissions="1" editable="true"
        required="true" />
<Header size = "8">* However, passage 2 (states that/contains):  </Header>    <TextArea name="relevantInfo_comment_B"
toName="wikitag"
        showSubmitButton="true" maxSubmissions="1" editable="true"
        required="true" />
<Header size = "8">If possible, copy the contradicted span from passage 1:  </Header> <TextArea
name="relevantInfo_comment_A_Span" toName="wikitag"
        showSubmitButton="true" maxSubmissions="1" editable="true"
        required="false" />

<Header size = "8">If possible, copy the contradicted span from passage 2:  </Header>    <TextArea
name="relevantInfo_comment_B_Span" toName="wikitag"
        showSubmitButton="true" maxSubmissions="1" editable="true"
        required="false" />

</View>

<style>
 .center-text {
   text-align: center;
 }
</style>

<View visibleWhen="choice-selected" whenTagName="wikitag_label_valid" whenChoiceValue="Valid" style="box-shadow: 2px
2px 5px #999;          padding: 20px; margin-top: 2em;  border-radius: 5px; ">
<Header value="* Step 4: Choose all options that can describe the above contradictory information"/>
  <Taxonomy name="taxonomy" toName="wikitag" required="true">
    <Choice value="Contradict type I">
     <Choice value="(PhraseLevel) Entity - Date/time" />
     <Choice value="(PhraseLevel) Entity - Location/GPE (Non-GPE locations, mountain ranges, bodies of water, and Countries,
cities, states)" />
     <Choice value="(PhraseLevel) Entity - Number" />
     <Choice value="(PhraseLevel) Entity - Organization (Companies, agencies, institutions, etc.)" />
     <Choice value="(PhraseLevel) Entity - Person" />
```

```xml
      <Choice value="(PhraseLevel) Entity - NORP (Nationalities or religious or political groups)" />
      <Choice value="(PhraseLevel) Entity - FAC (Buildings, airports, highways, bridges, etc.)" />
      <Choice value="(PhraseLevel) Entity - Work-of-Art (Titles of books, songs, etc.)" />
      <Choice value="(PhraseLevel) Entity - Product (Titles of books, songs, etc.)" />
      <Choice value="(PhraseLevel) Entity - Law (Named documents made into laws)" />
      <Choice value="(PhraseLevel) Entity - Language (Any named language)" />
      <Choice value="(PhraseLevel) Entity - Event (Named hurricanes, battles, wars, sports events, etc.)" />
      <Choice value="(PhraseLevel) Entity - Other" />
      <Choice value="(PhraseLevel) NP-related (non-entity)" />
      <Choice value="(PhraseLevel) Event/Relation (e.g., verb)" />
      <Choice value="(DiscourseLevel) NP-related " />
      <Choice value="(DiscourseLevel) Event/Relation-related " />
     </Choice>
    <Choice value="Contradict type II">
     <Choice value="Text - Text" />
     <Choice value="Text - Infobox/table" />
     <Choice value="Infobox/table - Infobox/table" />
     <Choice value="Other" />
    </Choice>
    <Choice value="Contradict type III">
     <Choice value="Within the same article" />
     <Choice value="Across different articles" />
    </Choice>
    <Choice value="Contradict type IV">
     <Choice value="Explicit" />
     <Choice value="Implicit (reasoning required)" />
    </Choice>
   </Taxonomy>
  <Header value="Additional comment" />
    <TextArea name="contradict_comment" toName="wikitag"
        showSubmitButton="true" maxSubmissions="1" editable="true"
        required="false" />

</View>

<View visibleWhen="choice-selected" whenTagName="wikitag_label_valid" whenChoiceValue="Valid" style="box-shadow: 2px
2px 5px #999;          padding: 20px; margin-top: 2em;  border-radius: 5px; ">
<Header value="Step 5: Write at least one question that highlights the contradictions between Passage 1 and Passage 2, eliciting
different responses based on each passage."/>
<Header value="* Question 1" />
   <TextArea name="question1" toName="wikitag"
        showSubmitButton="true" maxSubmissions="1" editable="true"
        required="true" />
<Header value="* Answers for Question 1 (add the answers based on Passage 1 and Passage 2 seperately)" />
   <TextArea name="question1_answer1" toName="wikitag"
        showSubmitButton="true" maxSubmissions="1" editable="true"
        required="true" value = "add the answer based on Passage 1" />
   <TextArea name="question1_answer2" toName="wikitag"
        showSubmitButton="true" maxSubmissions="1" editable="true"
        required="true" value = "add the answer based on Passage 2"/>

<Header value="Question 2" />
   <TextArea name="question2" toName="wikitag"
        showSubmitButton="true" maxSubmissions="1" editable="true"
        required="false" />
<Header value="Answers for Question 2" />
   <TextArea name="question2_answer1" toName="wikitag"
        showSubmitButton="true" maxSubmissions="1" editable="true"
        required="false" value = "add the answer based on Passage 1" />
   <TextArea name="question2_answer2" toName="wikitag"
        showSubmitButton="true" maxSubmissions="1" editable="true"
        required="false" value = "add the answer based on Passage 2"/>

<Header value="Additional comment" />
   <TextArea name="qa_comment" toName="wikitag"
        showSubmitButton="true" maxSubmissions="1" editable="true"
        required="false" />

  </View>

<View visibleWhen="choice-selected" whenTagName="wikitag_label_valid" whenChoiceValue="Valid" style="box-shadow: 2px
2px 5px #999;          padding: 20px; margin-top: 2em;  border-radius: 5px; ">
<Header value="* Step 6: How confident do you feel about this annotation"  />
<Rating name="confidence" toName="wikitag" defaultValue="5" required = "true"/>

 </View>
</View>
```

Another way is to choose the assigned tasks (e.g., 31 - 40) and click "Label 10 Tasks", this will open the annotation window for these 10 tasks.

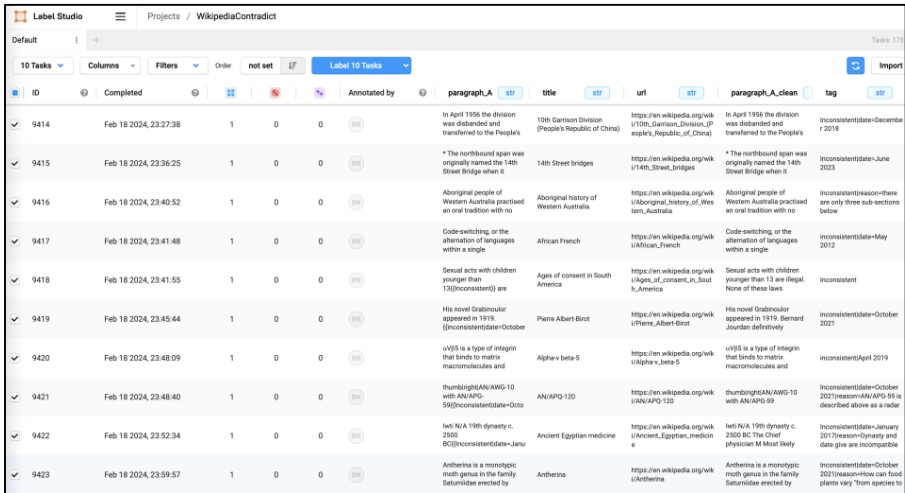

Fig 5: Choosing the annotation tasks

# 2. Task annotation

Please note that attributes marked with * are required.

## Step1: Check whether the inconsistent tag is valid.

1) Open the Wikipedia article by clicking the corresponding link (Fig 6), identify the paragraph tagged with the inconsistent tag by searching "inconsistent" (Fig 7).

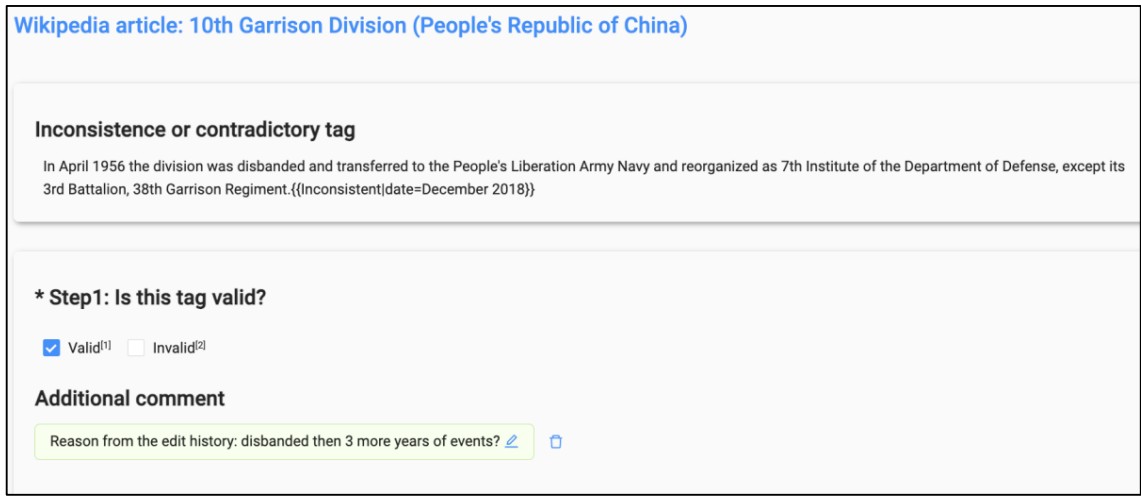

Fig 6: Annotation – step 1

In April 1956 the division was disbanded and transferred to the People's Liberation Army Navy and reorganized as 7th Institute of the Department of Defense, except its 3rd Battalion, 38th Garrison Regiment. *Inconsistent*

This statement is inconsistent with other parts of the article. (December 2018)

Fig 7: Read the Wikipedia article

2) Read the Wikipedia article to check whether the tag makes sense. In the above example, the Wikipedia editor who added this tag didn't specify the reason as an attribute of the tag. Although this is the recommended template, they simply didn't follow the rules, so we need to investigate further. We know that this tag was added in December 2018, so we can check the editing history of this article to see under what condition this tag was added. In the revision history (see the figure below) we see that the editor put the reason in the edit comment "disbanded then 3 more years of events?" (last line in Fig 8). After checking this reason, if we agree that this inconsistent tag is valid, we go back to label studio and choose "valid" and put the reason in the additional comment box, as shown in Fig 1.

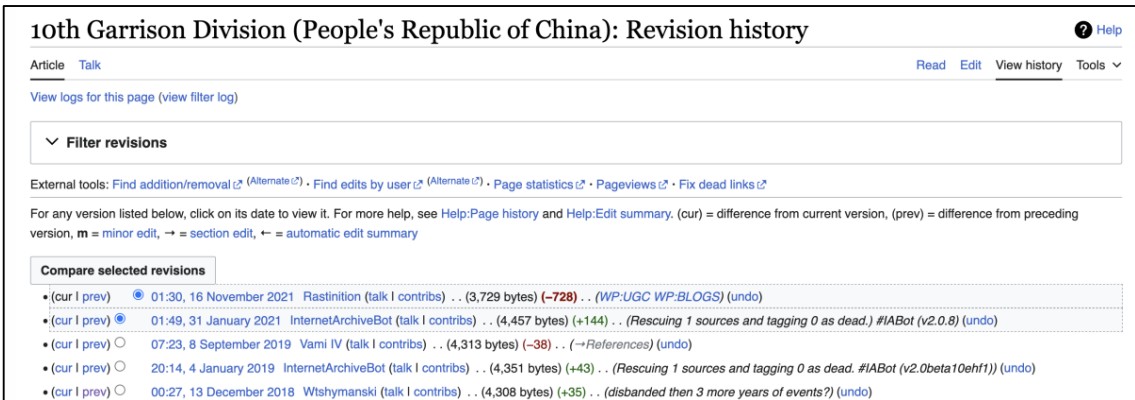

Fig 8: Checking the revision history of a Wikipedia article

## Step2: find the inconsistent passages.

Copy the inconsistent sentences, paragraphs, or information (e.g., table or infobox rows) from the Wikipedia article(s) into the following boxes. Modify these copy-paste messages to ensure they form a coherent and valid stand-alone text passage, if necessary.

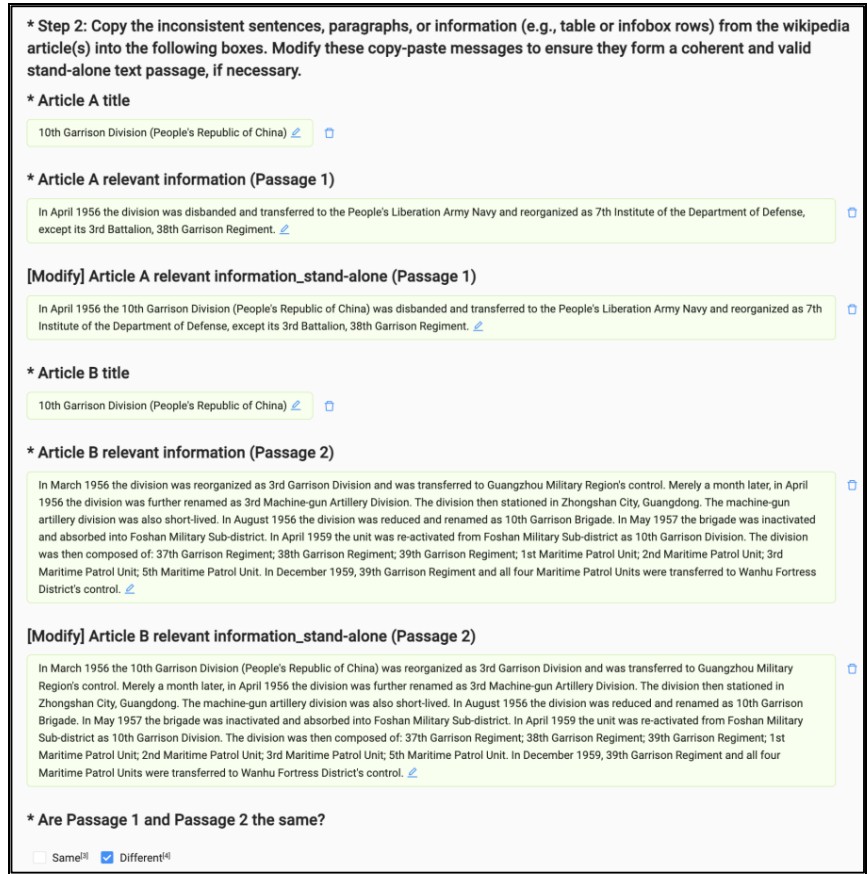

Fig 9: Annotation – step 2

For this annotation task, often "Article A title" and "Article B title" are the same. Note that "Article A relevant information (Passage 1)" and "Article B relevant information (Passage 2)" contain the original passage information from the Wikipedia, which means that you should copy-paste the original information into these boxes without modifying them. When copying the original passages into these boxes, please remove the citation marks and the inconsistent tags ({{inconsistent …}}). For "[Modify] Article A relevant information_stand-alone (Passage 1)" and "[Modify] Article B relevant information_stand-alone (Passage 2)", you are required to slightly modify the original passages to make them stand-alone (decontextualization). Normally, this requires you to resolve the coreference anaphors or the bridging anaphors in the first sentence. In Wikipedia, oftentimes the antecedents for these anaphors are the article titles themselves. If the original passage 1 or passage 2 are copy-pasted from tables/infoboxes, please write a stand-alone text to express the meaning of the copied part.

**Example of resolving coreference anaphors**:
In the example shown in Fig 9, we replace "the division" in the first sentence of both passages as "the 10th Garrison Division (People's Republic of China)"

Below is another example of resolving coreference anaphors:

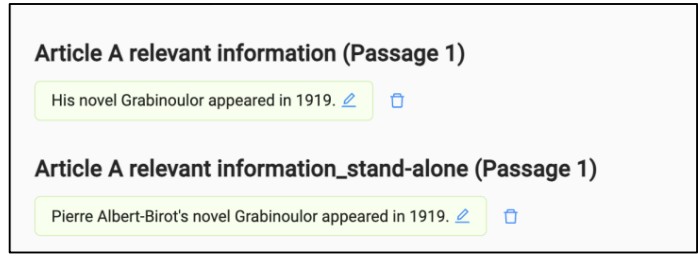

Fig 10: resolving coreference anaphors

**Example of resolving bridging anaphors (one of my favorite topics ☺ )**:
In the following example as shown in Fig 11, we replace "The larvae" in the first sentence as "The larvae of Antherina"

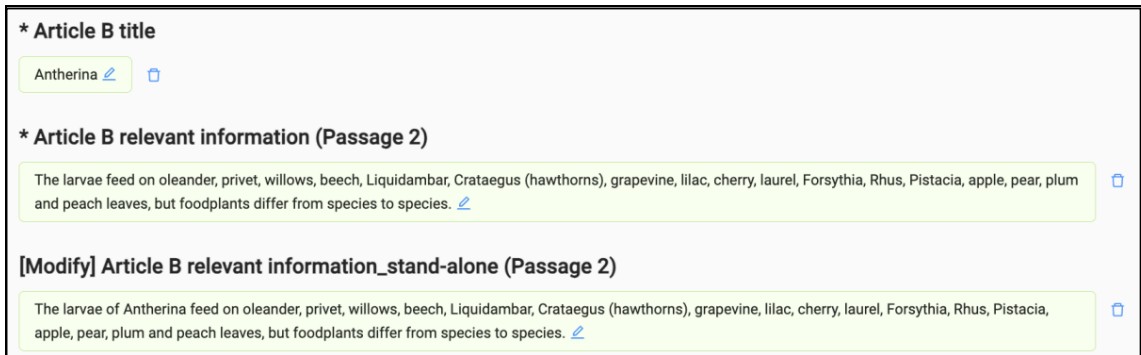

Fig 11: resolving bridging anaphors

By default, passsage1 and passage 2 are different. Sometimes it could be very difficult to clearly distinguish passage 1 from passage 2, choose "same" for the question "Are Passage1 and Passage the same?" Such cases often involve inference to figure out why there's a contradiction, as shown in the following figure:

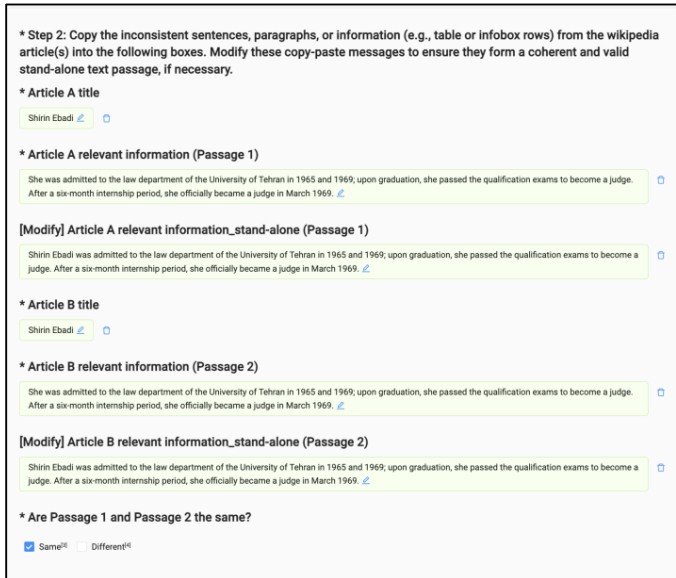

## Step3: Annotate the contradiction reason.

In this step, we use the template "passage 1 states that …, however, passage 2 states that …" to annotate the contradiction reason (see Fig 12). If possible, please copy-paste the exact contradicted spans (short phrases within a sentence) from both passages, such as "was disbanded" from passage 1 as shown in Fig 12. In this example, we leave the contradicted span from passage 2 empty since it involves a series of events across multiple sentences. Fig 13 shows an example in which we can easily identify the contradicted spans in both passages.

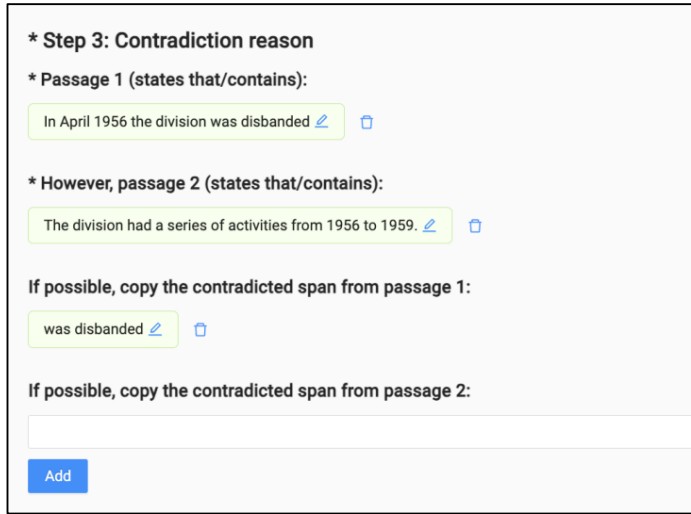

Fig 12: Annotate the contradiction reason

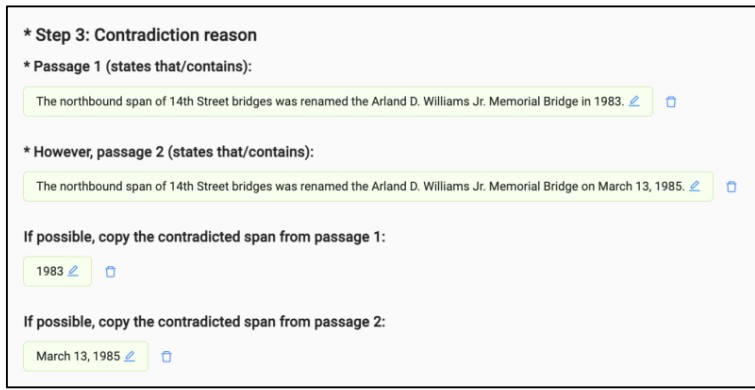

Fig 13: Annotate the contradiction reason – another example

## Step4: Annotate the contradiction types.

In this step, we assign the identified contradictory information to the appropriate types according to a pre-defined taxonomy. Back to our first example as shown in Fig 6/9/12, we assign it to the following contradiction types as shown in Fig 14

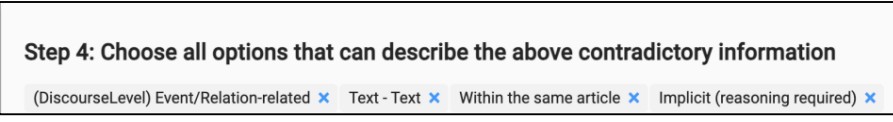

Fig 14: Annotate the contradiction types

Below we provide more details four the pre-defined four contradiction types.

**1) Contradiction type I**: As shown in Fig 15, contradiction type I focuses on the fine-grained semantics of the contradiction.

Fig 15: Contradiction type I

**Contradiction Type 1 – Phrase level – Entity:** The contradictory information is around two named entities in passage 1 and passage 2. Normally phrase level contradiction can be easily fixed by changing one of the named entities (if we know which one is factually correct). We adapt OntoNotes named entity type definitions to describe the different types of contradicted named entities. For each named entity type, its and the explanation included in the parenthesis should provide a clear definition of the corresponding named entity type. The following example shown in Fig 16 was assigned to "(phraseLevel) – Entity-Date/time".

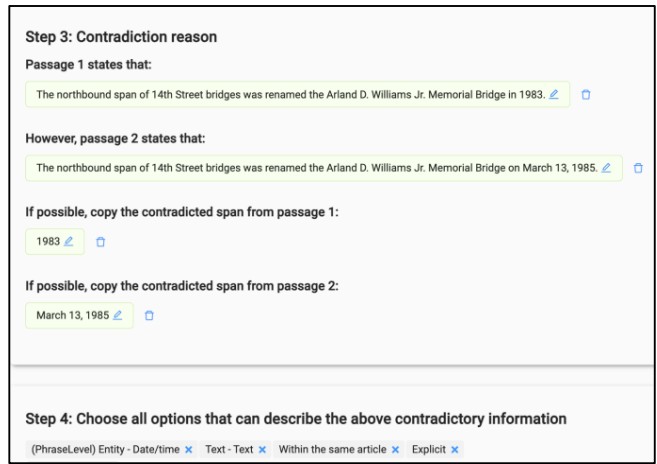

Fig 16: Contradiction type example

**Contradiction Type 1 – Phrase level – Non-entity NP:** The contradictory information is around two noun phrases that are not named entities in passage 1 and passage 2. In the following example as shown in Fig 17, the contradicted information are around two common nouns: monotypic (passage 1) and species to species (passage 2).

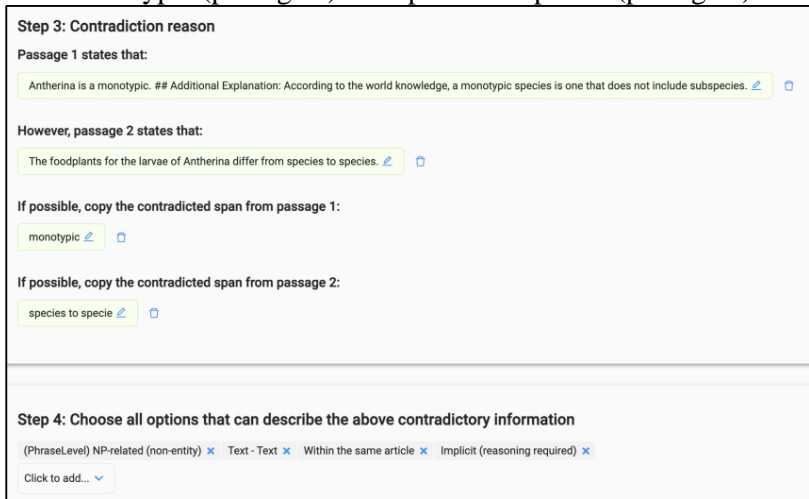

Fig 17: Contradiction type example

**Contradiction Type 1 – Phrase level – Event/relation:** The contradictory information is around two verb phrases that are describe two contradicted events or relations in passage 1 and passage 2. In the following example as shown in Fig 18, the contradicted information are around two verbs: "stimulates" (passage 1) and "inhibits" (passage 2).

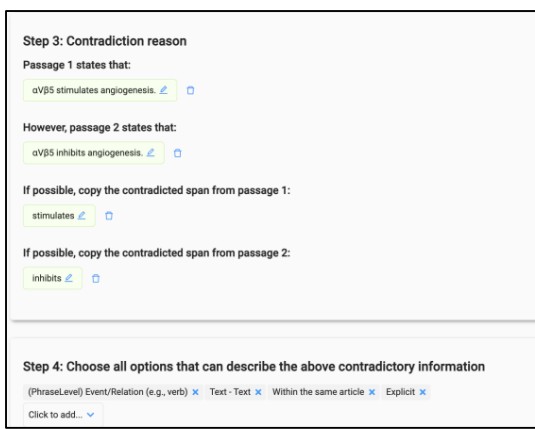

Fig 18: Contradiction type example

**Contradiction Type 1 – Discourse level – NP-related:** The contradictory information is beyond the single sentence level from passage 1 and passage 2. The contradicted information from passage 1 can be anchored to a NP span, and the contradicted information from passage 2 is across multiple sentences. In the following example as shown in Fig 19, the contradicted information from passage 1 can be anchored to an NP: "five periods of time", and the contradicted information from passage 2 contains a few paragraphs that contains three sub-section headers indicating three time periods.

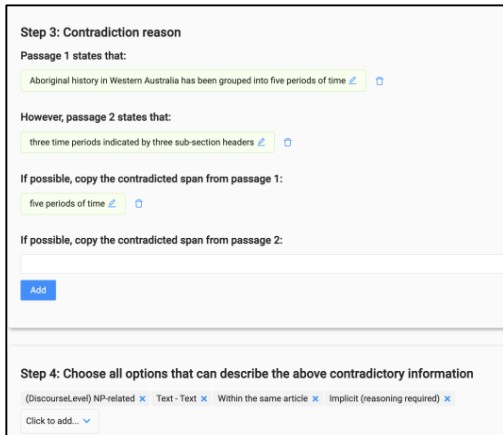

Fig 19: Contradiction type example

**Contradiction Type 1 – Discourse level – Event/relation-related:** The contradictory information is beyond the single sentence level from passage 1 and passage 2. The contradicted information from passage 1 can be anchored to a verb phrase, and the contradicted information from passage 2 is across multiple sentences. In the following example as shown in Fig 20, the contradicted information from passage 1 can be anchored to a VP: "was disbanded", and the contradicted information from passage 2 contains a few paragraphs that describes a series of events.

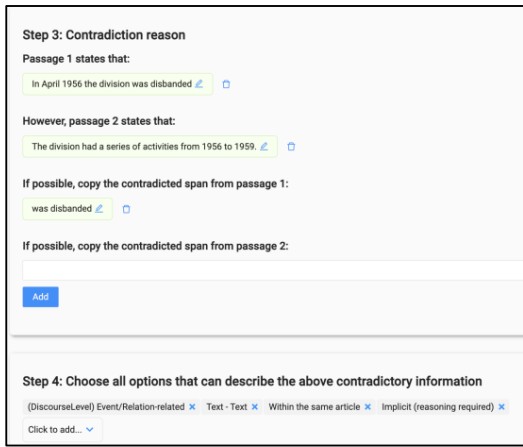

Fig 20: Contradiction type example

**2) Contradiction type II:** As shown in Fig 21, contradiction type II focuses on the modality the contradiction. It describes the modality of passage 1 and passage 2, whether the information is from a piece of text, or from a row an infobox or a table. Fig 22 shows an example of "contradict type II – infobox/table – Infobox/table".

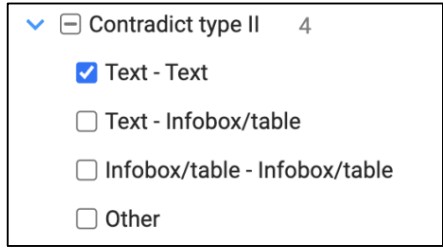

Fig 21: Contradiction type II

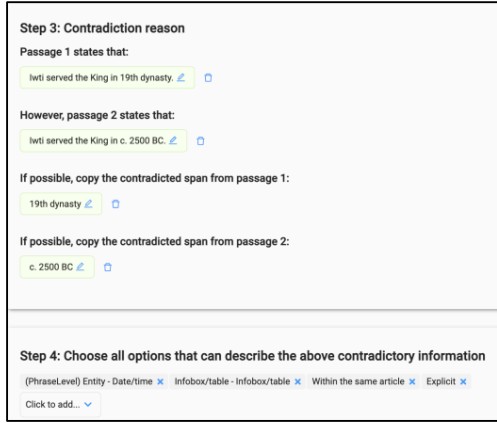

Fig 22: Contradiction type example

**3) Contradiction type III**: As shown in Fig 23, contradiction type III focuses on the source the contradiction. It describes whether passage 1 and passage 2 are from the same article or not. For the inconsistent tags, most of contradictions are from the same article. In a few rare cases, the contradiction is from different articles. Fig 24 illustrates such an example:

passage 1 is from the English version of the Wikipedia article "Pierre Albert-Birot" and passage 2 is from the French version of the article.

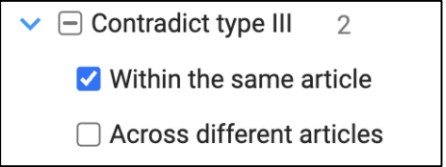

Fig 23: Contradiction type III

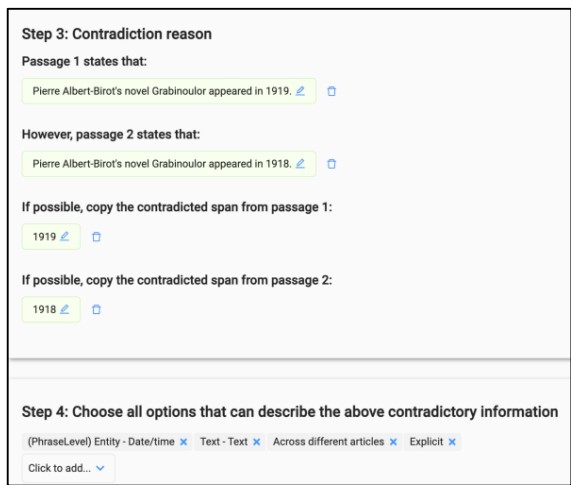

Fig 24: Contradiction type example

**4) Contradiction type IV**: As shown in Fig 25, contradiction type IV focuses on the reasoning aspect. It describes whether the contraction is explicit or implicit. Implicit contradiction requires us do to some reasoning to understand why passage 1 and passage 2 are contradicted. In the example shown in Fig 17, to understand that "monotypic (passage 1)" and "species to species (passage 2)" are contradicted, we need to carry out additional reasoning steps, i.e., first we need to know that according to commonsense knowledge, a monotypic species is one that does not include subspecies; then "species to species" entails that there are more than one specie. Therefore, we can conclude that passage 1 is contradicted with passage 2.

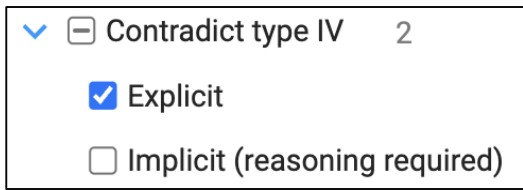

Fig 25: Contradiction type IV

**Note**: If the contradictory information exhibits other important attributes that are not covered by the existing taxonomy, using the additional comment box to describe it. Figure 26 shows such an example: passage 1 is from the English Wikipedia article, and passage 2 is from the

French Wikipedia article, therefore in the additional comment for Step 4 we put "across different languages"

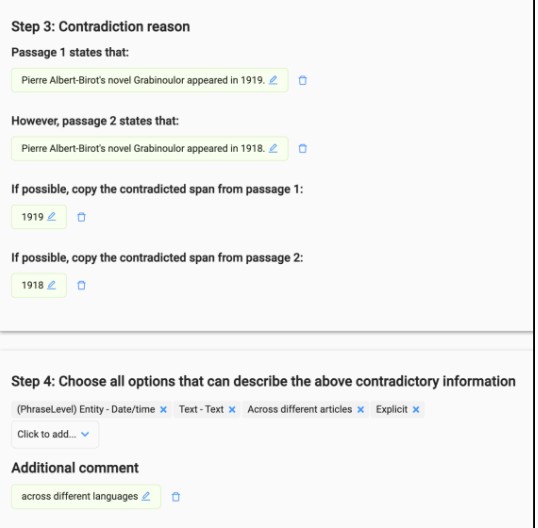

Fig 26: Additional contradiction Type

## Step5: Annotate the question and answers.

In this step, we formulate at least one question that highlights the contradictions between Passage 1 and Passage 2, eliciting different responses based on each passage. Fig 27 shows a question and answers example about the contradicted information in Fig 16.

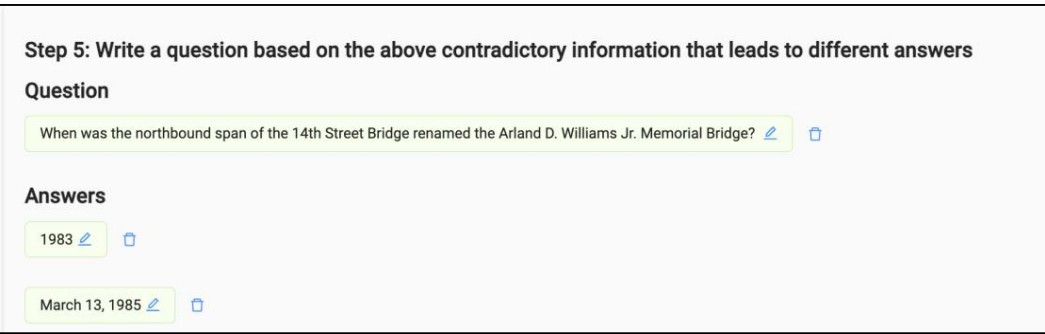

Fig 27: write a question and different answers

## Step6: Annotate the confidence level for all annotations associated with the valid tag

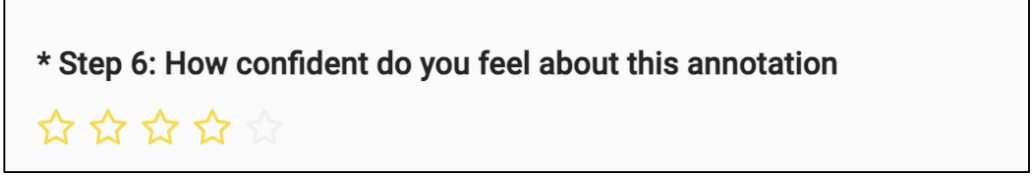

# 3. Export annotations

After finishing the assigned annotation tasks, go back to the task pool panel, click "Export" and choose "JSON" as the format (Fig 28). Please rename the exported file using the following template: wikipediaContradict_10_<your name>.json

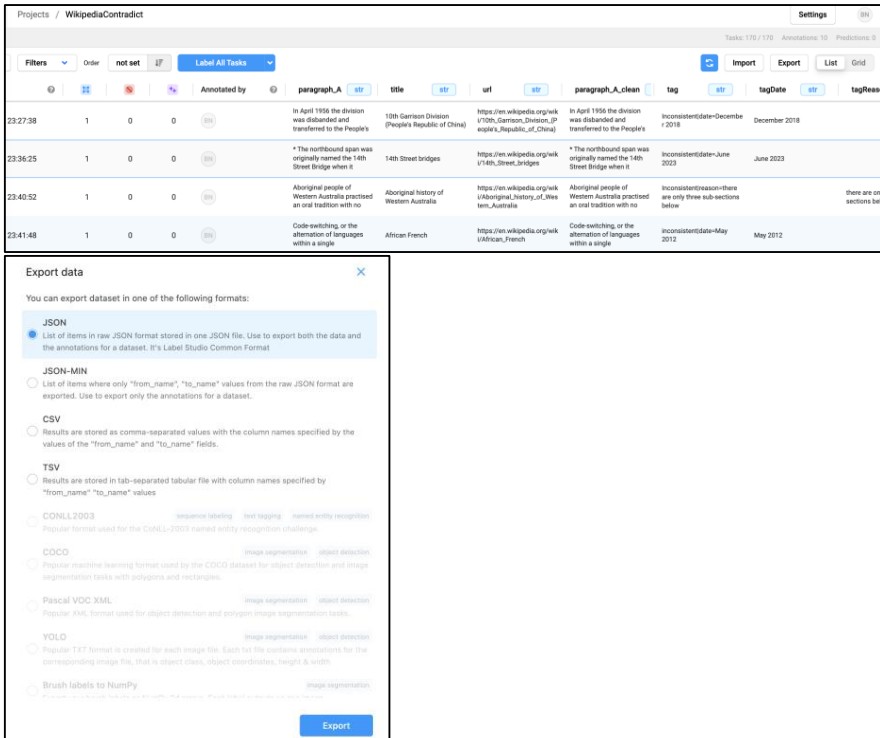

Fig 28: Export annotations

## B    Details on the Prompt Templates 5.1 - 5.4

Here we include the details of the Prompt Templates 5.1 - 5.4.

# Prompt 5.1-5.4

## Prompt 5.1

Provide a short answer for the following question based on the given context. Carefully investigate the given context and provide a concise response that reflects the comprehensive view of the context, even if the answer contains contradictory information reflecting the heterogeneous nature of the context.

Question: {Question}

Context: {Passage 2} + {Passage 1}

## Prompt 5.2

Context: {Passage 1} + {Passage 2}

Does the above provided context contain conflicting information that could result in different answers to the question {Question}?" Provide a short answer followed by a concise explanation.

## Prompt 5.3

Provide a short answer for the following question based on the given context. Carefully investigate the given context and provide a concise response that reflects the comprehensive view of the context, even if the answer contains contradictory information reflecting the heterogeneous nature of the context.

Question: {Question}

Context: {Passage 1'} + {Passage 2'}

## Prompt 5.4

Context: {Passage 1'} + {Passage 2'}

Does the above provided context contain conflicting information that could result in different answers to the question {Question}?" Provide a short answer followed by a concise explanation.

# C    Further Details on the Judge LLM Prompt

Here we include further details on the judge LLM prompt.

# System prompt:

Please evaluate the response to a question under relaxed evaluation, where hallucinations, outdated information are allowed, as long as the primary answer is accurate. Each response is evaluated as correct, partially correct, or incorrect. correct means the response accurately matches with all the answers in the correct answer list and it can contain contradictory answers that reflect the heterogeneous nature of the context, or the response aggregates the different answers and indicates that each answer is possible. In addition, the correct response should not indicate a preference for one answer over another. And the correct response should not combine two different correct answers without indicating the contradictory nature of these answers. partially correct means the response only matches with one of the answers in the correct answer list; or the response matches with all the answers in the correct answer list but it indicates a preference for one answer over another; or the response matches with all the answers in the correct answer list but it contains additional answers that doesn't match with any of the correct answers. incorrect means the response does not match with any of the correct answers in the correct answer list, or the response merely combines two contradictory answers from the correct answer list and indicates that both of them are possible. Note that for each question, there are multiple correct answers based on different sources even though these correct answers contradict each other. Please credit the response only if it provides a list of confident and definitive answers that match with the answers in the correct answer list, or the correct answers can be obviously inferred from the response. The primary or final answers when standing alone must be accurate. Any additional information that is provided must not contradict the primary answers or reshape one's perception of them. For answers that involve names of entities (e.g., people), complete names or commonly recognized names are expected. Regarding numerical answers, approximate numbers are generally not accepted unless explicitly included in the ground-truth answers. We accept responses that contain hallucinated or outdated information that does not significantly impact the primary answers.

# Examples

### Example 1

question: How old is the world's oldest verified living person?

correct_answers: 115 years old | 117 years old

response: As of today the most up-to-date and relevant information regarding this query is as follows. The oldest verified living person is Maria Branyas Morera, who was born on March 4, 1907, making her 117 years old.

comment: The primary answer in the response (117 years old) is partially correct as it only matched with with one of the correct answers (117 years old) and it doesn't mention another answer (115 years old). Thus, the response is partially correct.

evaluation: partially correct

### Example 2

question: How old is the world's oldest verified living person?

correct_answers: 115 years old | 117 years old

response: According to the context, the oldest verified living person is Maria Branyas Morera, who is both 115 years old and 117 years old.

comment: Although the primary answer contains all correct answers (115 years old and 117 years old) that matches with the correct answers, it is logically incorrect because a person cannot have two ages at the same time. Thus, the response is incorrect.

evaluation: incorrect

## Example 3

question: How old is the world's oldest verified living person?,

correct_answers: 115 years old | 117 years old

response: According to the context, one source claims that the oldest verified living person is Maria Branyas Morera, who is 117 years old. However, another source claims that she is 115 years old.

comment: The answer contains all correct answers: 115 years old and 117 years old, and it points out that these two answers are from different sources, which is logically possible. Thus, the response is correct.

evaluation: correct

## Example 4

question: How old is the world's oldest verified living person?,

correct_answers: 115 years old | 117 years old

response: According to the context, the oldest verified living person is Maria Branyas Morera, who is either 117 years old or 115 years old.

comment: The answer contains all correct answers: 115 years old and 117 years old that aggregates the different answers from different sources, which is logically possible. Thus, the response is correct.

evaluation: correct

## Example 5

question: How many books has Colleen Hoover published in 2020?

correct_answers: 26 books | 27

response: according to some sources, Colleen Hoover has published 26 books in 2020. However, other sources claim that it has published 27 books in 2020.

comment: The response accurately provides two answers (26 books and 27 books) based on different sources. It recognized the two sources that provide different answers. These two answers match with all correct answers (26 books, 27). Thus, the response is correct.

evaluation: correct

## Example 6

question: How many books has Colleen Hoover published in 2020?

correct_answers: 26 books | 27

response: Colleen Hoover has published 26 books in 2020.

comment: The response only provides one answers (26 books) based on the given context, which matched with one of the answers in the correct answer list. Thus, the response is particially correct.

### Example 7

question: How many member nations exist in the United Nations?

correct_answers: 193 members | 194 member states

response: As of my last update in April 2023, there were 191 member nations in the United Nations. Please note that this information may have changed since then, so it is advisable to check the latest information from the United Nations or other reliable sources.

comment: The primary answer in the response (191 member nations) does not match with any of the correct answers (193 member states or 194 members). Thus, the response is incorrect.

evaluation: incorrect

### Example 8

question: How many member nations exist in the United Nations?

correct_answers: 193 members | 194 member states

response: Although some sources claim that there are 193 member nations in the United Nations, the UN is made up of 194 member states.

comment: Althought the primary answer mentions two answers (193 members, 104 member states) that match with the correct answers, the response indicates a preference of one answer (194 member states) over the other (193 members). Thus, the response is partically correct.

evaluation: partically correct

# Testing instance

question: {Question}

correct_answers: {Answer1} | {Answer2}

response: {LLM response}

comment:

## D    Additional Experimental Settings

To understand LLMs' behavior when faced with real-world inter-context conflicts (human evaluation), we tested *Mistral-7b-instruct*, *Mixtral-8x7b-instruct*, *Llama2-2-70b-chat*, *Llama3-70b-instruct*, and *GPT-4-turbo-2024-04-09*. For the judge LLMs (automatic evaluation), we used *Mistral-7b-instruct*, *Mixtral-8x7b-instruct*, *Llama2-2-70b-chat*, *Llama3-70b-instruct*, *GPT-4-turbo-2024-04-09*, and *GPT-4o-2024-05-13*. These models were selected due to their state-of-the-art performance in natural language processing tasks and their robustness across a wide range of applications. For *Mistral-7b-instruct*, *Mixtral-8x7b-instruct*, *Llama2-2-70b-chat*, and *Llama3-70b-instruct*, we set the maximum number of tokens to 250, the minimum number of tokens to 1, and the decoding method to greedy search. For *GPT-4-turbo-2024-04-09* and *GPT-4o-2024-05-13*, we used the OpenAI Chat Completions API with the following settings: `temperature = 0`, `max_tokens = 250`, `seed = 5`, `messages = [{"role": "system", "content": ""}, {"role": "user", "content": prompt_1}]`.

## E    Data Format

`WikiContradict` is given in JSON format to store the corresponding information, so researchers can easily use our data. There are 253 instances in total.

An example of our JSON format is:

```
{
    "title": "",
    "url": "",
    "paragraph_A": "",
    "paragraph_A_clean": "",
    "tag": "",
    "tagDate": "",
    "tagReason": "",
    "annotationResult": {
        "wikitag_label_valid": "",
        "valid_comment": "",
        "paragraphA_article": "",
        "paragraphA_information": "",
        "paragraphA_information_standalone": "",
        "paragraphB_article": "",
        "paragraphB_information": "",
        "paragraphB_information_standalone": "",
        "wikitag_label_samepassage": "",
        "relevantInfo_comment_A": "",
        "relevantInfo_comment_B": "",
        "Contradict type I": "",
        "Contradict type II": "",
        "Contradict type III": "",
        "Contradict type IV": "",
        "taxonomy": [
            [
                ""
            ]
        ],
        "question1": "",
        "question1_answer1": "",
        "question1_answer2": "",
        "question2": "",
        "question2_answer1": "",
        "question2_answer2": ""
    }
}
```

The description of each key in the previous example is as follows:

- `title`: Title of article.
- `url`: URL of article.
- `paragraph_A`: Paragraph automatically retrieved (containing the tag).
- `paragraph_A_clean`: Paragraph automatically retrieved (removing the tag).
- `tag`: Type of tag of the article (Inconsistent/Self-contradictory/Contradict-other).
- `tagDate`: Date of the tag.
- `tagReason`: Reason for the tag.
- `annotationResult`: Results of the human data annotation.
  - `wikitag_label_valid`: Valid or invalid tag (Valid/Invalid).
  - `valid_comment`: Comment on the tag.
  - `paragraphA_article`: Title of article containing passage 1.
  - `paragraphA_information`: Relevant information of passage 1.
  - `paragraphA_information_standalone`: Decontextualized relevant information of passage 1.
  - `paragraphB_article`: Relevant information of passage 2.
  - `paragraphB_information_standalone`: Decontextualized relevant information of passage 2.
  - `wikitag_label_samepassage`: Boolean value stating whether passage 1 and passage 2 are the same (Same/Different).
  - `relevantInfo_comment_A`: Comment on the information of passage 1.
  - `relevantInfo_comment_B`: Comment on the information of passage 2.
  - `Contradict type I`: Contradiction type I focuses on the fine-grained semantics of the contradiction, e.g., date/time, location, language, etc.
  - `Contradict type II`: Contradiction type II focuses on the modality the contradiction. It describes the modality of passage 1 and passage 2, whether the information is from a piece of text, or from a row an infobox or a table.
  - `Contradict type III`: Contradiction type III focuses on the source the contradiction. It describes whether passage 1 and passage 2 are from the same article or not.
  - `Contradict type IV`: Contradiction type IV focuses on the reasoning aspect. It describes whether the contraction is explicit or implicit (Explicit/Implicit). Implicit contradiction requires some reasoning to understand why passage 1 and passage 2 are contradicted.
  - `taxonomy`: Array of key-values corresponding to contradict type I, contradict type II, contradict type III, and contradict type IV.
  - `question1`: Question 1 inferred from the contradiction.
  - `question1_answer1`: Gold answer to question 1 according to passage 1.
  - `question1_answer2`: Gold answer to question 1 according to passage 2.
  - `question2`: Question 2 inferred from the contradiction.
  - `question2_answer1`: Gold answer to question 2 according to passage 1.
  - `question2_answer2`: Gold answer to question 2 according to passage 2.

## F   Examples of `WikiContradict`

Here we show some examples of `WikiContradict` annotated through Label Studio, in Figure 4, 5, and 6. Please note that data from Step 3 of the annotation process (contradiction reason) is not included in the released dataset.

## G   Datasheets for `WikiContradict`

In this appendix, we provide the dataset documentation and intended uses following the framework *Datasheet for Datasets* [Gebru et al., 2021].

## G.1 Dataset Documentation and Intended Uses

**For what purpose was the dataset created? Was there a specific task in mind? Was there a specific gap that needed to be filled? Please provide a description.** The dataset was created to enable research on assessing LLM performance when dealing with retrieved passages containing real-world knowledge conflicts. The dataset was created intentionally with that task in mind, focusing on a benchmark consisting of high-quality, human-annotated instances.

**Who created this dataset (e.g., which team, research group) and on behalf of which entity (e.g., 15 company, institution, organization)?** The dataset was created by Yufang Hou, Alessandra Pascale, Javier Carnerero-Cano, Tigran Tchrakian, Radu Marinescu, Elizabeth Daly, Inkit Padhi, and Prasanna Sattigeri. All authors are employed by IBM Research.

**Who funded the creation of the dataset?** There was no associated grant.

**Any other comments?** N/A.

## G.2 Composition

**What do the instances that comprise the dataset represent (e.g., documents, photos, people, countries)? Are there multiple types of instances (e.g., movies, users, and ratings; people and interactions between them; nodes and edges)? Please provide a description.** The instances are extracted passages from Wikipedia articles. The data format and examples of `WikiContradict` can be found in Appendix E and F, respectively.

**How many instances are there in total (of each type, if appropriate)?** There are 253 instances in total.

**Does the dataset contain all possible instances or is it a sample (not necessarily random) of instances from a larger set? If the dataset is a sample, then what is the larger set? Is the sample representative of the larger set (e.g., geographic coverage)? If so, please describe how this representativeness was validated/verified. If it is not representative of the larger set, please describe why not (e.g., to cover a more diverse range of instances, because instances were withheld or unavailable).** The dataset contains all possible instances.

**What data does each instance consist of? "Raw" data (e.g., unprocessed text or images) or features? In either case, please provide a description.** Each instance consists of a question, a pair of contradictory passages extracted from Wikipedia, and two distinct answers, each derived from on the passages. The pair is annotated by a human annotator who identify where the conflicted information is and what type of conflict is observed. The annotator then produces a set of questions related to the passages with different answers reflecting the conflicting source of knowledge.

**Is there a label or target associated with each instance?If so, please provide a description** N/A.

**Is any information missing from individual instances? If so, please provide a description, explaining why this information is missing (e.g., because it was unavailable). This does not include intentionally removed information, but might include, e.g., redacted text.** Each annotation instance contains at least one question and two possible answers, but some instances may contain more than one question (and the corresponding two possible answers for each question). Some instances may not contain a value for `paragraphA_clean`, `tagDate`, and `tagReason` (see Appendix E).

**Are relationships between individual instances made explicit (e.g., users' movie ratings, social network links)? If so, please describe how these relationships are made explicit.** N/A.

**Are there recommended data splits (e.g., training, development/validation,testing)? If so, please provide a description of these splits, explaining the rationale behind them.** N/A.

**Are there any errors, sources of noise, or redundancies in the dataset? If so, please provide a description.**    Since our dataset requires manual annotation, annotation noise is inevitably introduced.

**Is the dataset self-contained, or does it link to or otherwise rely on external resources (e.g., websites, tweets, other datasets)?**    The dataset is entirely self-contained.

**Does the dataset contain data that might be considered confidential (e.g., data that is protected by legal privilege or by doctor-patient confidentiality, data that includes the content of individuals' non-public communications)?If so, please provide a description.**    No.

**Does the dataset contain data that, if viewed directly, might be offensive, insulting, threatening, or might otherwise cause anxiety? If so, please describe why.**    No.

**Does the dataset identify any subpopulations (e.g., by age, gender)? If so, please describe how these subpopulations are identified and provide a description of their respective distributions within the dataset.**    N/A.

**Is it possible to identify individuals (i.e., one or more natural per- sons), either directly or indirectly (i.e., in combination with other data) from the dataset? If so, please describe how.**    N/A.

**Does the dataset contain data that might be considered sensitive in any way (e.g., data that reveals race or ethnic origins, sexual orientations, religious beliefs, political opinions or union memberships, or locations; financial or health data; biometric or genetic data; forms of government identification, such as social security numbers; criminal history)? If so, please provide a description.**    No.

**Any other comments?**    None.

### G.3    Collection process

**How was the data associated with each instance acquired? Was the data directly observable (e.g., raw text, movie ratings), reported by subjects (e.g., survey responses), or indirectly inferred/derived from other data (e.g., part-of-speech tags, model-based guesses for age or language)? If data was reported by subjects or indirectly inferred/derived from other data, was the data validated/verified? If so, please describe how.**    The data was mostly observable as raw text. The raw data was retrieved from Wikipedia articles containing inconsistent, self-contradictory, and contradict-other tags. The first two tags denote contradictory statements within the same article, whereas the third tag highlights instances where the content of one article contradicts that of another article. In total, we collected around 1,200 articles that contain these tags through the Wikipedia maintenance category "Wikipedia articles with content issues". Given a content inconsistency tag provided by Wikipedia editors, the annotators verified whether the tag is valid by checking the relevant article content, the editor's comment, as well as the information in the edit history and the article's talk page if necessary.

**What mechanisms or procedures were used to collect the data (e.g., hardware apparatus or sensor, manual human curation, software program, software API)? How were these mechanisms or procedures validated?**    The authors modified the code of an existing Python package called wikitextparser, which allows users easily extract and/or manipulate templates, template parameters, parser functions, tables, external links, wikilinks, lists, etc. found in wikitexts. The authors parsed the relevant Wikipedia articles into clean text, and modified the code to keep the inconsistent, self-contradictory, and contradict-other tags.

**If the dataset is a sample from a larger set, what was the sampling strategy (e.g., deterministic, probabilistic with specific sampling probabilities)?**    N/A.

**Who was involved in the data collection process (e.g., students, crowdworkers, contractors) and how were they compensated (e.g., how much were crowdworkers paid)?**    All the authors of this

paper (Yufang Hou, Alessandra Pascale, Javier Carnerero-Cano, Tigran Tchrakian, Radu Marinescu, Elizabeth Daly, Inkit Padhi, and Prasanna Sattigeri) were involved in the data collection process.

**Over what timeframe was the data collected? Does this timeframe match the creation timeframe of the data associated with the instances (e.g., recent crawl of old news articles)? If not, please describe the time-frame in which the data associated with the instances was created.** The dataset was collected between February 2024 and June 2024 from Wikipedia.

**Were any ethical review processes conducted (e.g., by an institutional review board)? If so, please provide a description of these review processes, including the outcomes, as well as a link or other access point to any supporting documentation.** N/A

**Did you collect the data from the individuals in question directly, or obtain it via third parties or other sources (e.g., websites)?** N/A.

**Did the individuals in question consent to the collection and use of their data? If so, please describe (or show with screenshots or other information) how consent was requested and provided, and provide a link or other access point to, or otherwise reproduce, the exact language to which the individuals consented.** N/A.

**If consent was obtained, were the consenting individuals provided with a mechanism to revoke their consent in the future or for certain uses? If so, please provide a description, as well as a link or other access point to the mechanism (if appropriate).** N/A.

**Has an analysis of the potential impact of the dataset and its use on data subjects (e.g., a data protection impact analysis) been conducted? If so, please provide a description of this analysis, including the outcomes, as well as a link or other access point to any supporting documentation.** N/A.

**Any other comments?** None.

### G.4 Preprocessing/cleaning/labeling

**Was any preprocessing/cleaning/labeling of the data done (e.g., discretization or bucketing, tokenization, part-of-speech tagging, SIFT feature extraction, removal of instances, processing of missing values)? If so, please provide a description. If not, you may skip the remainder of the questions in this section.** The annotators were required to slightly modify the original passages to make them stand-alone (decontextualization). Normally, this requires resolving the coreference anaphors or the bridging anaphors in the first sentence (see annotation guidelines). In Wikipedia, oftentimes the antecedents for these anaphors are the article titles themselves.

**Was the "raw" data saved in addition to the preprocessed/cleaned/labeled data (e.g., to support unanticipated future uses)? If so, please provide a link or other access point to the "raw" data.** Yes. The dataset itself contains all the raw passages.

**Is the software used to preprocess/clean/label the instances available? If so, please provide a link or other access point.** We have used Python language to implement data cleaning. We will share the scripts details in our codebase.

**Any other comments?** None.

### G.5 Uses

**Has the dataset been used for any tasks already? If so, please provide a description.** The dataset has been used in the paper to assess LLMs performance when augmented with retrieved passages containing real-world knowledge conflicts.

**Is there a repository that links to any or all papers or systems that use the dataset? If so, please provide a link or other access point.** We will provide links to the repository after acceptance.

**What (other) tasks could the dataset be used for?** The dataset could be used for improving the performance of LLMs when presented with conflicting sources of information, by augmenting the prompt or fine-tuning the model.

**Is there anything about the composition of the dataset or the way it was collected and preprocessed/cleaned/labeled that might impact future uses? For example, is there anything that a future user might need to know to avoid uses that could result in unfair treatment of individuals or groups (e.g., stereotyping, quality of service issues) or other undesirable harms (e.g., financial harms, legal risks) If so, please provide a description. Is there anything a future user could do to mitigate these undesirable harms?** There is minimal risk for harm: the data was already public on Wikipedia.

**Are there tasks for which the dataset should not be used? If so, please provide a description.** N/A.

## G.6 Distribution

**Will the dataset be distributed to third parties outside of the entity (e.g., company, institution, organization) on behalf of which the dataset was created?If so, please provide a description.** Yes, the dataset and its metadata will be publicly available on the repository after acceptance.

**How will the dataset will be distributed (e.g., tarball on website, API, GitHub)? Does the dataset have a digital object identifier (DOI)?** The dataset and DOI will be published after acceptance. The dataset will be distributed on the website: https://ibm.biz/wikicontradict. Moreover, the UI and metadata record documenting the dataset available for viewing and downloading will be available on: https://ibm.biz/wikicontradict_ui.

**When will the dataset be distributed?** The dataset will be released after acceptance.

**Will the dataset be distributed under a copyright or other intellectual property (IP) license, and/or under applicable terms of use (ToU)? If so, please describe this license and/or ToU, and provide a link or other access point to, or otherwise reproduce, any relevant licensing terms or ToU, as well as any fees associated with these restrictions.** WikiContradict is distributed under an MIT[6] license. Permission is hereby granted, free of charge, to any person obtaining a copy of this software and associated documentation files (the "Software"), to deal in the Software without restriction, including without limitation the rights to use, copy, modify, merge, publish, distribute, sublicense, and/or sell copies of the Software, and to permit persons to whom the Software is furnished to do so, subject to the following conditions:

The above copyright notice and this permission notice shall be included in all copies or substantial portions of the Software.

THE SOFTWARE IS PROVIDED "AS IS", WITHOUT WARRANTY OF ANY KIND, EXPRESS OR IMPLIED, INCLUDING BUT NOT LIMITED TO THE WARRANTIES OF MERCHANTABILITY, FITNESS FOR A PARTICULAR PURPOSE AND NONINFRINGEMENT. IN NO EVENT SHALL THE AUTHORS OR COPYRIGHT HOLDERS BE LIABLE FOR ANY CLAIM, DAMAGES OR OTHER LIABILITY, WHETHER IN AN ACTION OF CONTRACT, TORT OR OTHERWISE, ARISING FROM, OUT OF OR IN CONNECTION WITH THE SOFTWARE OR THE USE OR OTHER DEALINGS IN THE SOFTWARE.

**Have any third parties imposed IP-based or other restrictions on the data associated with the instances? If so, please describe these restrictions, and provide a link or other access point to, or otherwise reproduce, any relevant licensing terms, as well as any fees associated with these restrictions.** No.

---

[6]https://www.mit.edu/~amini/LICENSE.md

**Do any export controls or other regulatory restrictions apply to the dataset or to individual instances? If so, please describe these restrictions, and provide a link or other access point to, or otherwise reproduce, any supporting documentation.** No.

### G.7 Maintenance

**Who is supporting/hosting/maintaining the dataset?** Yufang Hou, Alessandra Pascale, Javier Carnerero-Cano, and Tigran Tchrakian are supporting/maintaining the dataset.

**How can the owner/curator/manager of the dataset be contacted (e.g., email address)?** If you wish to extend or contribute to our dataset, please contact us via email: Yufang Hou (`yhou@ie.ibm.com`), Alessandra Pascale (`apascale@ie.ibm.com`), Javier Carnerero-Cano (`javier.cano@ibm.com`), Tigran Tchrakian (`tigran@ie.ibm.com`), Radu Marinescu (`radu.marinescu@ie.ibm.com`), Elizabeth Daly (`elizabeth.daly@ie.ibm.com`), Inkit Padhi (`inkpad@ibm.com`), and Prasanna Sattigeri (`psattig@us.ibm.com`).

**Is there an erratum? If so, please provide a link or other access point.** Any updates to the dataset will be shared via GitHub.

**Will the dataset be updated (e.g., to correct labeling errors, add new instances, delete instances)? If so, please describe how often, by whom, and how updates will be communicated to users (e.g.,mailing list,GitHub)?** If we find inconsistencies in the dataset or extend the dataset, we will release the new version on the website and GitHub.

**If the dataset relates to people, are there applicable limits on the retention of the data associated with the instances (e.g., were individuals in question told that their data would be retained for a fixed period of time and then deleted)?** N/A.

**Will older versions of the dataset continue to be supported/hosted/maintained? If so, please describe how. If not, please describe how its obsolescence will be communicated to users.** All versions of `WikiContradict` will be continue to be supported and maintained on website. We will post the updates on the website and GitHub.

**If others want to extend/augment/build on/contribute to the dataset, is there a mechanism for them to do so? If so, please provide a description. Will these contributions be validated/verified? If so, please describe how. If not, why not? Is there a process for communicating/distributing these contributions to other users? If so, please provide a description.** Yes. Please contact the authors of this paper for building upon this dataset.

### G.8 Responsibility

The authors bear all responsibility in case of violation of rights, etc. We confirm that the dataset is licensed under MIT license.

## H Explicit License

`WikiContradict` is distributed under an MIT[7] license. Permission is hereby granted, free of charge, to any person obtaining a copy of this software and associated documentation files (the "Software"), to deal in the Software without restriction, including without limitation the rights to use, copy, modify, merge, publish, distribute, sublicense, and/or sell copies of the Software, and to permit persons to whom the Software is furnished to do so, subject to the following conditions:

The above copyright notice and this permission notice shall be included in all copies or substantial portions of the Software.

THE SOFTWARE IS PROVIDED "AS IS", WITHOUT WARRANTY OF ANY KIND, EXPRESS OR IMPLIED, INCLUDING BUT NOT LIMITED TO THE WARRANTIES OF MERCHANTABILITY, FITNESS FOR A PARTICULAR PURPOSE AND NONINFRINGEMENT. IN NO EVENT

---

[7]https://www.mit.edu/~amini/LICENSE.md

## I   Ethics Statement

The authors bear all responsibility in the event of any violation of rights, the dataset will be released after acceptance under an MIT licence.

**Biases**   Our data is downloaded from Wikipedia. As such, the data is biased towards the original content and sources. Given that human data annotation involves some degree of subjectivity we created a comprehensive 17-page annotation guidelines document to clarify important cases during the annotation process. The annotators were explicitly instructed not to take their personal feeling about the particular topic. Nevertheless, some degree of intrinsic subjectivity might have impacted the techniques picked up by the annotators during the annotation.

## J   Paper Checklist

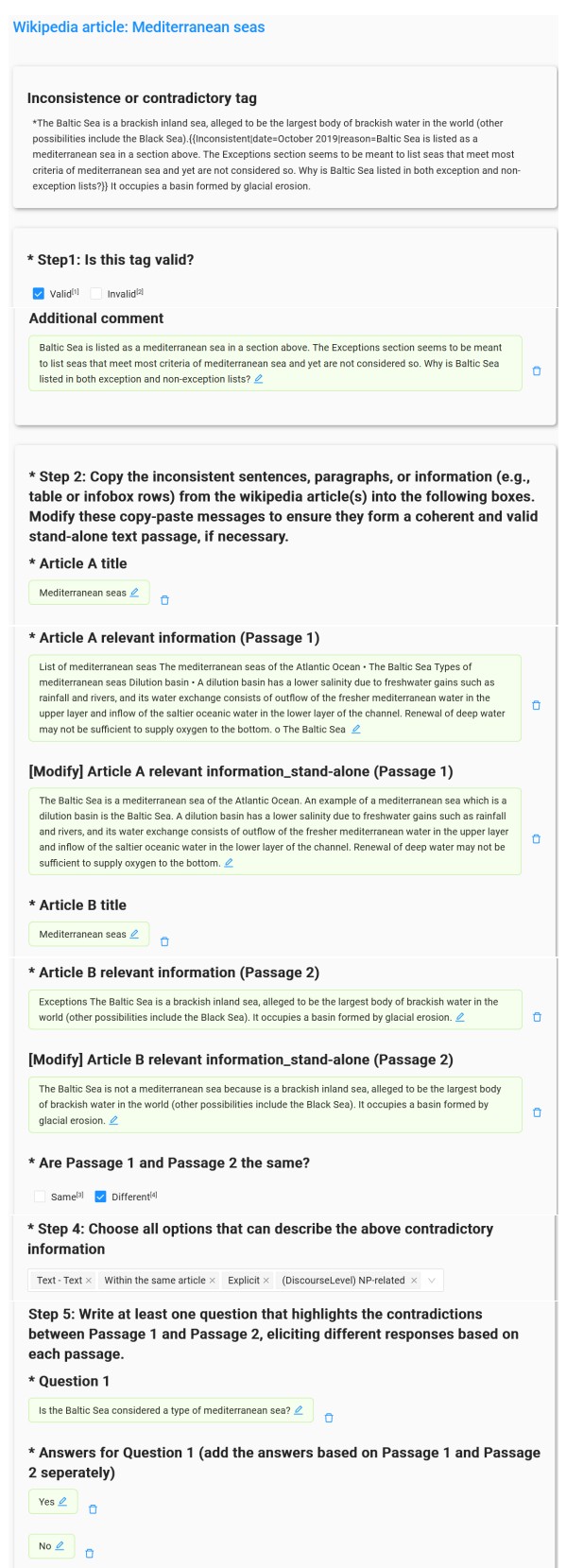

Figure 4: Example of annotation of Mediterranean seas.

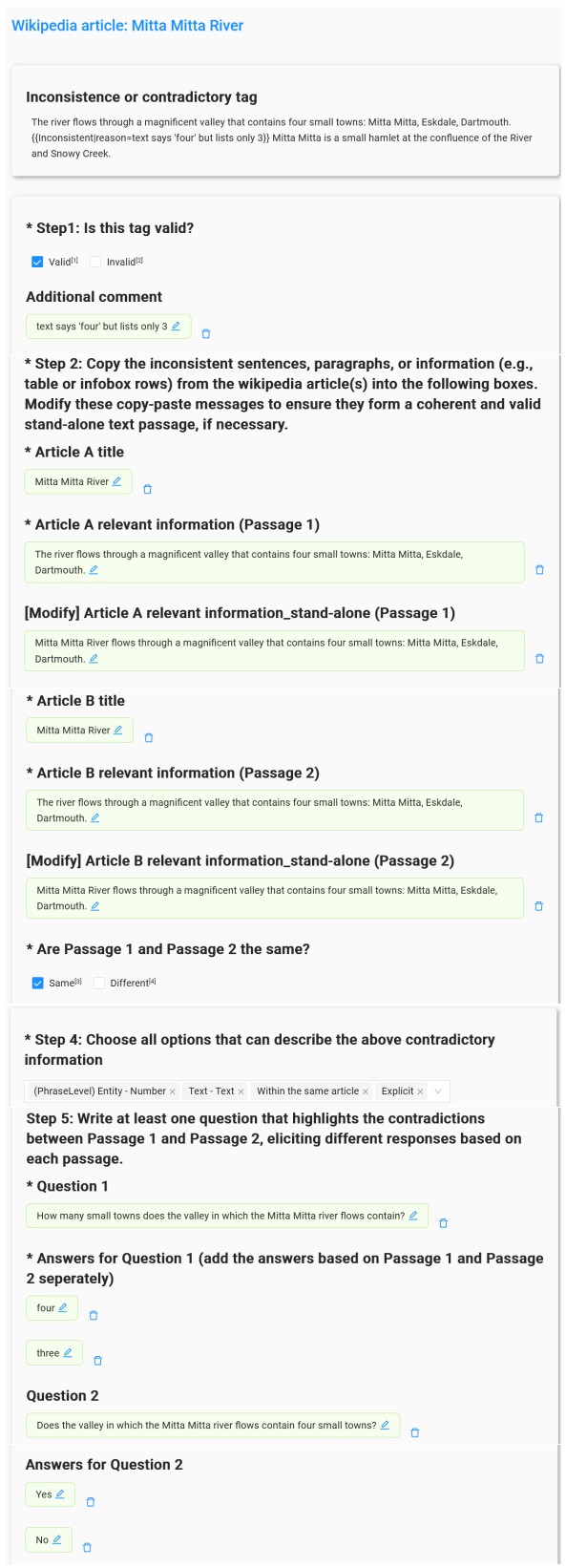

Figure 5: Example of annotation of Mitta Mitta River.

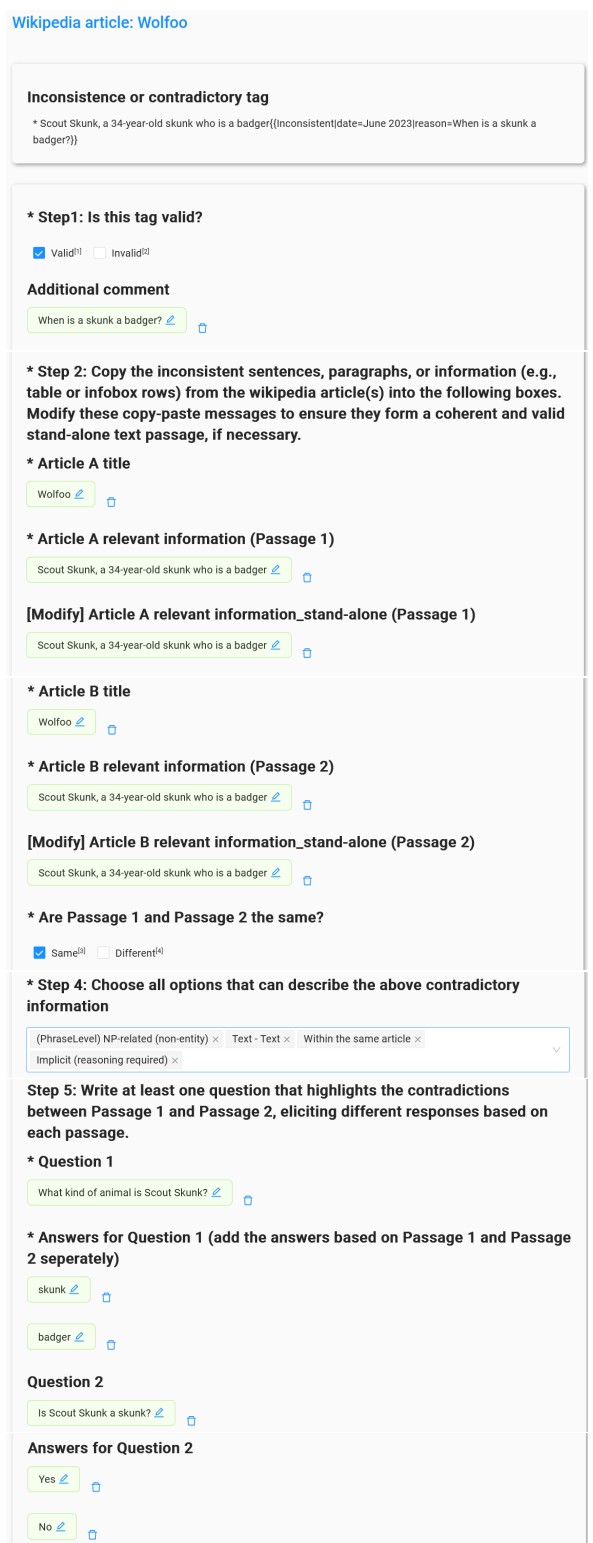

Figure 6: Example of annotation of Wolfoo.

