# OpenReview forum: "WikiContradict: A Benchmark for Evaluating LLMs on Real-World Knowledge Conflicts from Wikipedia"
_NeurIPS.cc/2024/Datasets_and_Benchmarks_Track — NeurIPS 2024 Track Datasets and Benchmarks Poster_

### Official Review · Reviewer_t17j · 2024-06-21
**A good RAG benchmark with contradicted knowledge**

**Rating:** 6
**Confidence:** 4

**Review:**

This paper addresses an important and timely problem in RAG evaluation - how models handle conflicting information from credible sources. The WikiContradict dataset provides carefully curated examples of real-world contradictions from Wikipedia.

The evaluation methodology is thorough, testing models under multiple conditions and using both human and automated evaluation.
The results provide interesting insights into current LLM limitations, showing that models struggle to properly handle contradictory information even when explicitly prompted to do so. The analysis of explicit vs implicit contradictions is particularly illuminating and it should spur further research into improving how LLMs reason about and reconcile conflicting information.

Overall, this is a solid contribution that addresses an important gap in current LLM evaluation.

**Strengths:**

- Novel dataset derived from real-world Wikipedia contradictions through an insightful approach that utilizes inconsistency tags
- Comprehensive evaluation of multiple LLMs under different conditions
- Rigorous human evaluation methodology
- Insightful analysis of results, particularly around explicit vs implicit contradictions

**Additional Feedback:**

N/A

**Clarity:**

The paper is generally well-written and clearly structured. Supplementary materials provide detailed information for interested readers.

**Correctness:**

The methodology appears sound and well-executed. The human evaluation process is rigorous, with good inter-annotator agreement.

**Documentation:**

The paper and supplementary materials provide very detailed steps to reproduce the efforts.

**Ethics:**

No concern about the ethics.

**Limitations:**

- As suggested by the authors, reliance on Wikipedia tags may bias the types of contradictions covered
- The data provided in the attachment is still noisy as the passages that are supposed to be contraditory sometimes are not.
- How to properly handle contradiction hasn't been carefully discussed. Simply outlining the contradiction in the response seems to be a good approach, but is this always the case? What if some typos are made that can be easily resolved?

**Opportunities For Improvement:**

- The dataset mostly contains public information that has been seen at pre-training stage and possibly from various websites. It is hard to force the model to avoid using intrinsic knowledge and stick to context without careful prompting. Therefore it will be better for the authors to evaluate answer factuality together with the proposed metrics.
- The paper studies the challenge of handling contradicted information in the context in zero-shot, but doesn't study the few-shot setups where examples of expected behavior of handling contradictions is provided in the instruction.
- The paper could benefit from more discussion of potential approaches to improve LLM performance on this task2

**Relation To Prior Work:**

Not familar with this area. But it seems this paper does a good job of positioning this work relative to prior benchmarks and evaluations of LLM factuality and contradiction handling.

**Summary And Contributions:**

This paper introduces WikiContradict, a new benchmark dataset mined from Wikipedia for evaluating how LLMs handle real-world knowledge conflicts. The key contributions are:

1. A novel dataset of 253 human-annotated instances derived from Wikipedia articles with contradictory information.
2. A comprehensive evaluation of multiple LLMs on this dataset under different QA scenarios.
3. Human evaluation results on a subset of the data, revealing limitations of current LLMs in handling contradictory information.
4. An automated evaluation method (WikiContradictEval) to facilitate large-scale evaluations.

---

> ### Author Rebuttal · Authors · 2024-08-28
>
> We thank Reviewer t17j for the constructive feedback. We appreciate that the reviewer pointed out that our work “addresses an important and timely problem in RAG evaluation - how models handle conflicting information from credible sources” and “the evaluation methodology is thorough”.
>
> Below we provide a detailed response to the reviewer’s comments:
>
> **1. About the data contamination issue:** since Wikipedia is widely regarded as a credible and high-quality resource by almost all LLMs, LLMs have likely encountered most of these contradictions during their pre-training stages. We agree that it’s very challenging to instruct LLMs to avoid using their intrinsic knowledge and rely only with the given context --- in particular, we observe that GPT-4 exhibits a “stubborn” behavior, particularly with prompt template 3. It often relies on its internal knowledge, which may not align with the given context. In terms of “factuality”, if it means whether an LLM answer can be grounded in the given context, we think that our “incorrect” category in all tables partially provides such information, as it means that a response does not match any of the annotated answers, or the response merely combines two contradictory answers from the annotated answer list and indicates that both are possible at the same time without indicating the contradictory nature of the two context passages. We appreciate the suggestion of combining our evaluation metric with the answer factuality evaluation. In the finally version, we will add an additional evaluation metric (with the name of factuality or groundedness) in Table 2 by assessing whether the response can be grounded in the given context.
>
> **2. About the few-shot experiments and potential approaches to improve LLMs on this task:** we thank the reviewer for pointing out this aspect. This work focuses on evaluating LLMs in handling real-world conflicts in the RAG scenario, therefore we carried out all experiments in the zero-shot setup. In the on-going follow-up work, we are designing new approaches to improve LLMs’ performance on this task, including prompting LLMs with a few examples as an intuitive baseline. We will provide a short discussion on future research directions for improving LLMs' performance on this task in the final version.
>
> **3. Regarding the noise in the provided supplementary material:** we appreciate the reviewer's diligence in reviewing our work. If any issues are identified in the annotated dataset, please let us know, and we will correct them in the final version.
>
> **4. Regarding “How to properly handle contradiction”**: thanks for raising this question. We agree that the factual contradictions are relatively easy to resolve if the source of the contradiction is known, i.e., in which context such contradiction occurs in the first place (e.g., somebody made a typo when editing the name of a politician in a Wikipedia article). The problem is that often times we don’t know how and why such contradictions occur after the information was produced. In such cases, we believe that a good option is outlining the contradiction in the response given that all retrieved passages are from the same credible resource. An alternative approach would be to cross-check each answer separately with other credible resource(s) and provide a confidence score of the correctness of each individual answer. We will discuss this further in the final version.

---

### Official Review · Reviewer_A47E · 2024-07-24
**WikiContradict Review**

**Rating:** 7
**Confidence:** 4
**Correctness:** Yes
**Clarity:** Yes

**Review:**

Pros and Cons:

Pros:
- The work builds a nice dataset with real-world examples for testing LMs
- The evaluation covers a wide range of settings and LMs
- The paper is easy to read and well-written

Cons:
 - The evaluation seems a bit odd to me at first but I've come around - it was not super clear from the abstract/intro that the goal of the eval is to identify both answers in conflicting documents. Perhaps that's my bias from reading other papers on knowledge conflicts where the goal is usually to disambiguate why it prefers one over the other.
- The paper lacks some highly salient related work (see Refs) which includes even a work of almost the exact same title. I would've expected some more discussion on this topic.

Overall, I think the paper is a nice benchmark and provides interesting data.  Evaluation is difficult, auto-LM judging has its flaws as the authors show, and the task is somewhat different from what LMs are expecting, but I think it will be a nice start for new work on the topic.

Questions for the authors: is there any non-contradicting data given to the models? I think the current evaluation focuses on recall of the contradicting answers but doesn't handle the precision case - I wonder whether models would find contradictions that don't exist. This is minor.

**Strengths:**

The paper proposes an interesting evaluation and dataset to assess knowledge conflicts in LLMs with realistic data. Many of them are in fact highly challenging.

**Additional Feedback:**

N/A

**Documentation:**

I didn't see any in the paper but perhaps it is anonymized. I assume it will be publicly released.

**Limitations:**

Yes

**Opportunities For Improvement:**

Things I would suggest:
- More information about what the evaluation task is in the abstract and intro, it's not obvious you expect the models to identify and respond with both contradicting answers.
- Adding more discussion about related work on the topic

**Relation To Prior Work:**

References:
- WikiContradiction: Detecting Self-Contradiction Articles on Wikipedia seems to be a very similar motivation and approach (albeit older) using the Wikipedia tags. Also the dataset name is nearly identical...
- Rich Knowledge Sources Bring Complex Knowledge Conflicts: Recalibrating Models to Reflect Conflicting Evidence was one of (if not the first) to examine how to get LMs to identify when there are knowledge conflicts
- Defending Against Misinformation Attacks in Open-Domain Question Answering was one of the first (if not the first) to try to resolve knowledge conflicts of conflicting information

Otherwise, it does discuss some of the related work I expected to see.

**Summary And Contributions:**

This paper proposes to mine Wikipedia for tags that mark a lack of consistency and then use those to simulate various experiments with LLMs.  They collect around 1200 articles and design a prompting evaluation setup to see whether LMs can pull out both of the answers. They find mixed results with various LMs and also show that LMs can perform somewhat well as evaluating other LMs.

---

> ### Author Rebuttal · Authors · 2024-08-28
>
> We thank Reviewer A47E for the positive feedback about our work, pointing out the value of our annotated dataset and the comprehensive evaluation experiments. Below, we provide detailed answers to the questions:
>
> **1. Regarding the evaluation protocol:** thank you for the suggestion! Indeed, the goal of our evaluation task is different from the prior work on knowledge conflicts, in which we task LLMs to provide a complete perspective of conflicts from the retrieved documents instead of choosing one answer over another. We will add more information about our evaluation protocol in the abstract and introduction sections in the final version.
>
> **2. Regarding the missing references:** Thanks a lot for the suggestion. We will incorporate the suggested references in the camera-ready version.
>
> **3. Regarding experiments with non-contradicting data given to the models:** In the submission, we have carried out additional experiments to test LLMs by manually resolving the contradictions between passage 1 and passage 2 (line 243 – line 245). Due to space limitations, we condensed the description of these additional experiments and the corresponding results into a single paragraph (see “additional evaluations” on page 7). In general, we observe that all models demonstrate high performance in correctly answering questions in the RAG setup, in which there are no contradictions between passage 1 and passage 2. However, LLMs struggle to detect when there are no contradictions in the given context, and GPT-4 and Llama-3-70b-instruct frequently “make up” reasons to explain why the two passages contradict with each other. We will improve the layout of this section and move the human evaluation results in Appendix B about these experiments to the main content in the final version.
>
> **4. About documentation:** we provided the following documentation in a sperate supplement material: A) Appendix A – WikipediaContradictionAnnotationGuideline; B) Appendix B - Additional Human Evaluation; C) Appendix C - LLM Judge Prompt; D) metadata of the constructed dataset. All of them will be publicly released. Currently, the dataset itself and the metadata can be accessed at: https://huggingface.co/datasets/ibm/Wikipedia_contradict_benchmark

---

### Official Review · Reviewer_qfC4 · 2024-07-25
**real-world inter-context conflicts with comprehensive human labeling**

**Rating:** 9
**Confidence:** 3
**Correctness:** Most claims seem reasonably correct.
**Clarity:** The paper is clear and easy to follow.

**Review:**

The paper is clearly written and targetting an important and practical problem. The conclusion is very informative and some insights (e.g. it is useful to explicitly prompt for conflict awareness) are very valuable to future research and applications.

**Strengths:**

- Important problem and reasonable solution, clever usage of wiki editing metadata
- Comprehensive labeling standardization and significant human efforts in labeling
- Informative statistics and experiment interpretations

**Additional Feedback:**

N/A

**Documentation:**

Documentation on the data collection procedure with labeling tool is comprehensive.

**Limitations:**

While most limitations are discussed in the limitation section, it remains interesting that how much difference would the conclusion be if the dataset is not from manual wiki collection, but with simple entity replacement. How "realistic" helps in leading to different conclusions?

**Opportunities For Improvement:**

- Would be helpful to provider finer-grain categories for the contradition types (in addition to just labeling explicit/implicit)
- Helpful to discuss automatic evaluation with other models than GPT-4 (even though their performance might not be as good)
- Insights on error study would be helpful (more insights on specific error reasons for partially correct / incorrect)

**Relation To Prior Work:**

Related works are well-discussed.

**Summary And Contributions:**

The paper present a wiki-based dataset with factual conflicts for evaluating LLM's ability in handling conflicting facts. It describes a standardized labeling procedure as well as comprehensive experiments to show the usage of the dataset in many different settings.

---

> ### Author Rebuttal · Authors · 2024-08-28
>
> We would like to express our gratitude to Reviewer qfC4 for recognizing that our submission targets “an important and practical problem” and the insights provided in the paper are “very valuable to future research and applications”.
>
> Below we provide detailed explanations and action plans to address the reviewer’s feedback:
>
> **1.  About the fine-grained category of the contradiction types:** during the data annotation process, we designed a taxonomy to cover fine-grained contradiction types. Due to the space limitation, we only briefly mention that we use a pre-defined taxonomy to categorize contradiction types and provide detailed definitions of explicit contradicts and implicit contradicts (line 144 - 147) since in the paper we mainly focus on analyzing LLMs’ behaviour in handling these two contradict types. The details about this taxonomy and the definition of other fine-grained contradiction types can be can be found in the supplement materials (Appendix A – WikipediaContradictionAnnotationGuideline.docx;  page 11 – page 16). Briefly speaking, our contradiction taxonomy contains four dimensions:
>
> * A) Contradiction type I focuses on the fine-grained semantics of the contradiction. In the phrase level entity contradiction, we adapt OntoNotes named entity type definitions to describe the different types of contradicted named entities, such as Date/time, Location, Number, etc.
>
> * B) Contradiction type II focuses on the modality the contradiction. It describes the modality of passage 1 and passage 2, whether the information is from a piece of text, or from a row of an infobox or a table.
>
> * C) Contradiction type III describes whether the contradictions are from the same wikipedia article or not.
>
> * D) Contradiction type IV focuses on the reasoning aspect.  It describes whether the contraction is explicit or implicit. In the camera-ready version, we will add a short description about our fine-grained contradiction scheme and refer the readers to check the appendix for more details.
>
> **2. About discussing automatic evaluation with other models than GPT-4:** In Table 3, we compare the performance of different LLM judge model results, including GPT-4, Llama-3-70b-inst, and Mixtral-8x7b-inst. We further provide a detailed analysis of Llama-3-70b-inst in Table 4 by comparing human judgement with Llama3-70b-instruct judgement. In general, we observe that the performance of “weak” open-source models, such as Mixtral-8x7b-inst falls behind compared to GPT-4 when used as a judge model, as it tends to misclassify partially correct answers as correct. But strong open models such as Llama-3-70b-inst perform on par with GPT-4 with similar F-scores on the correct category. We will add more explanation about the weak judge model’s performance in the final version.
>
> **3. About more insights on specific error reasons for partially correct / incorrect:** Thanks for the suggestion! In the RAG setup (Table 2, prompt template 2 - 5), models’ answers rarely fall into the “incorrect” category. When such cases happen, the model’s answer acknowledge coexistence of two facts which are not logically correct, such as stating a person was born on two dates (e.g., According to the provided context, Paul McCole was born on 1 February 1972 and 10 February 1972). In contrast, when instructed to answer questions based on their internal knowledge without providing any context (Table 2, prompt template 1), the ratio of "incorrect" answers increases significantly. This is likely due to the fact that LLMs either memorize a different answer from another source other than Wikipedia during pre-training or hallucinate the answer. Regarding partially correct answers, most LLMs tend to produce a single answer based on only one given context, neglecting the other. However, some models, such as Mistral-7b-inst, attempt to reconcile the conflicting information by providing both answers and then explaining why one of them is incorrect. This phenomenon is particularly pronounced in the Mistral-7b-inst model.
>
> **4. Regarding a synthetic dataset (with entity replacement) vs. realistic dataset (a diverse range of conflict types from Wikipedia):** Thank you for raising this question, which is closely related to the motivation of our work. Our primary goal is to assess how well LLMs perform in dealing with real-world scenarios, rather than relying on synthetically created conflicts, as is the case in most prior work on knowledge conflicts. We agree that it would be very interesting to compare LLMs behaviour in these two setups by carefully controlling how we generate the synthetic dataset. For instance, replacing one contradicted entity in one passage with a new entity of the same type (ensuring that one entity doesn’t appear in the LLM pre-training stage), and replacing both contradicted entities on the two passages with two new entities of the same type (ensuring that both entities don’t appear in the LLM pre-training stage). We anticipate LLMs will perform better on such tasks when the conflicting statements are clearly and logically conflicting, however real-world conflicting statements can be more nuanced and therefore more challenging which was a key focus for this work. While comparing performances on these synthetically generated instances will be interesting, it is a complimentary avenue we can explore as future work and we can add this to the discussion.

---

### Decision · Program_Chairs · 2024-09-26

**Decision:**

Accept (Poster)

**Comment:**

This is the meta-review that summarizes the review comments and discussions. The paper presents a wiki-based dataset with factual conflicts for evaluating LLM's ability to handle conflicting facts. All reviewers agree that the paper is well-written and addresses an important problem. Using wiki editing metadata is a reasonable solution and the evaluation covers a wide range of settings and LMs. The human evaluation results reveal the limitations of LLMs in handling contradictory information, and the automated evaluation facilitates large-scale evaluations. The benchmark and insights on the results are valuable to future research.